# An automated 13.5 hour system for scalable diagnosis and acute management guidance for genetic diseases

Mallory J. Owen[1,2], Sebastien Lefebvre [3], Christian Hansen[1,2], Chris M. Kunard[4], David P. Dimmock [1,2,5], Laurie D. Smith[1], Gunter Scharer[1], Rebecca Mardach[2,6], Mary J. Willis[1], Annette Feigenbaum[2,6], Anna-Kaisa Niemi[2,6], Yan Ding[1,2], Luca Van Der Kraan[1,2], Katarzyna Ellsworth[1,2], Lucia Guidugli[1,2], Bryan R. Lajoie[4], Timothy K. McPhail[4], Shyamal S. Mehtalia[4], Kevin K. Chau[1,2], Yong H. Kwon[1,2], Zhanyang Zhu [1,2], Sergey Batalov [1,2], Shimul Chowdhury[1,2,5], Seema Rego [1,2], James Perry[2,6], Mark Speziale[2,6], Mark Nespeca[2,6,7], Meredith S. Wright[1,2,5], Martin G. Reese [8], Francisco M. De La Vega[8], Joe Azure[8], Erwin Frise [8], Charlene Son Rigby [8], Sandy White[8], Charlotte A. Hobbs[1,2,6], Sheldon Gilmer [2], Gail Knight[2,6], Albert Oriol[1,2], Jerica Lenberg[1,2,5], Shareef A. Nahas[1,2], Kate Perofsky[1,2,6], Kyu Kim[1,2,6], Jeanne Carroll[1,2,6], Nicole G. Coufal[1,2,6], Erica Sanford[1], Kristen Wigby[1,2,6], Jacqueline Weir[4], Vicki S. Thomson[4], Louise Fraser [4], Seka S. Lazare [4], Yoon H. Shin[4], Haiying Grunenwald[4], Richard Lee[4], David Jones [4], Duke Tran[4], Andrew Gross[4], Patrick Daigle[4], Anne Case[4], Marisa Lue[4], James A. Richardson[4], John Reynders [3], Thomas Defay [3], Kevin P. Hall [4], Narayanan Veeraraghavan[1,2] & Stephen F. Kingsmore [1,2,5✉]

While many genetic diseases have effective treatments, they frequently progress rapidly to severe morbidity or mortality if those treatments are not implemented immediately. Since front-line physicians frequently lack familiarity with these diseases, timely molecular diagnosis may not improve outcomes. Herein we describe Genome-to-Treatment, an automated, virtual system for genetic disease diagnosis and acute management guidance. Diagnosis is achieved in 13.5 h by expedited whole genome sequencing, with superior analytic performance for structural and copy number variants. An expert panel adjudicated the indications, contraindications, efficacy, and evidence-of-efficacy of 9911 drug, device, dietary, and surgical interventions for 563 severe, childhood, genetic diseases. The 421 (75%) diseases and 1527 (15%) effective interventions retained are integrated with 13 genetic disease information resources and appended to diagnostic reports (https://gtrx.radygenomiclab.com). This system provided correct diagnoses in four retrospectively and two prospectively tested infants. The Genome-to-Treatment system facilitates optimal outcomes in children with rapidly progressive genetic diseases.

A full list of author affiliations appears at the end of the paper.

Collectively, the 7200 known genetic disorders engender a large proportion of pediatric morbidity and mortality, particularly in neonatal, pediatric, and cardiovascular ICUs[1–7]. Of 140 million children worldwide suffering from rare genetic diseases, it is estimated ~30% will not survive to their fifth birthday[8,9]. In ICU settings, progression of childhood genetic diseases is often extremely rapid leading to morbidity and/or early death without a timely diagnosis and treatment[10–12]. An initial, comprehensive technological solution to this problem was rapid diagnostic whole genome sequencing (rWGS), which enabled concomitant diagnostic evaluation of almost all genetic diseases in as little as 19.5 h[13–18]. rWGS is now being implemented nationally for inpatient diagnosis of childhood genetic disease in England, Wales, Germany, in Medicaid beneficiaries in Michigan, California, Minnesota, and Oregon, and in Anthem/Blue Cross/Blue Shield beneficiaries nationwide (https://www.genomicsengland.co.uk/matt-hancock-announces-5-million-genomes-within-five-years/) (https://www.blueshieldca.com/bsca/bsc/public/common/PortalComponents/provider/StreamDocumentServlet?fileName=PRV_WholeExome_Sequen.pdf) (https://dejure.org/gesetze/SGB_V/64e.html) (https://files.medi-cal.ca.gov/pubsdoco/bulletins/artfull/ips202112.aspx) (https://www.michigan.gov/documents/mdhhs/MSA_21-33_732848_7.pdf).

As is often true in biotechnology, rWGS removed one bottleneck, but exposed another downstream—delayed, variable, or absent implementation of optimal, specific treatments. Clinical trials of rWGS have identified several factors that contribute to the gap between expected and observed clinical utility of genetic disease diagnoses:[3,4,7,10] Firstly, exponential advances in genomics have outpaced medical education (https://dejure.org/gesetze/SGB_V/64e.html). Most healthcare providers lack adequate genomic literacy to practice genomic medicine, and depend upon other subspecialists, particularly medical geneticists, for translation of genome reports into treatment recommendations[19–22]. Geographic distance to specialty centers correlates with time to diagnosis, receipt of specialty care, and outcomes in childhood genetic diseases[23,24]. In quaternary hospitals, subspecialty and superspecialty consultation leads to delays in optimal treatment. In front-line settings, lack of a full complement of subspecialists greatly limits the clinical utility of rWGS. Secondly, many genetic diseases were either discovered only recently, or are ultra-rare, and therefore evidence-based treatment guidelines have not yet been developed. Management strategies are often interspersed across the literature in the form of case reports, case series or small cohort studies, and their relative effectiveness may not have been adjudicated. Information resources pertaining to management of rare genetic diseases are incomplete, lack interoperability, and are typically not targeted toward acute ICU treatment or front-line physicians. Upon receipt of an rWGS-based diagnosis, these factors put an unsupportable burden on front-line physicians to search and synthesize the available treatment evidence for rare genetic diseases, many of which they may have never encountered previously. As genetic diseases are discovered, and effective, n-of-few, genetic therapies proliferate, therapeutic unfamiliarity and unwarranted variation in clinical practice will increase[25–27]. Thirdly, failure to order rWGS as a first-tier test frequently leads to diagnosis at time of hospital discharge, when management plans have been solidified or, for rapidly progressive diseases, too late to have full clinical utility[10].

Here, we describe a comprehensive, scalable, biotechnology solution to delayed molecular diagnosis and substandard therapy in rapidly progressive childhood genetic diseases. Called Genome-to-Treatment, it is an automated, virtual system for genetic disease diagnosis and acute management guidance.

## Results

**13.5-hour genome sequencing.** We previously described genetic disease diagnosis by rWGS in 19.5 h[16]. However, clinical usefulness was limited by lack of scalability and insensitivity for copy number variants (CNVs) or structural variants (SVs), which underpin 20% of genetic diagnoses in children in ICUs[14,16,18,28]. Inclusive of CNV and SV detection, turnaround time was >30 h, which was insufficient for the most rapidly progressive childhood genetic diseases, such as neonatal encephalopathies[10,14,18,28]. We re-engineered rWGS to improve scalability, turnaround time, analytic performance for CNVs and SVs, and generalization to other healthcare systems (Fig. 1).

First, we simplified ordering of rWGS. Orders are placed directly through the Epic EHR (Fig. 1). The test order and patient metadata is transferred from the EHR to a custom ordering portal. Second, we developed a simpler, faster method of sequencing library preparation that retained the capability to identify CNVs and SVs, using magnetic bead-linked transposomes (DNA polymerase chain reaction-free kit, Illumina) (https://www.illumina.com/content/dam/illumina-marketing/documents/products/appnotes/illumina-dna-pcr-free-wgs-app-note-770-2020-006.pdf)[29]. Incubation steps were maximally reduced from those in the manufacturer's protocol (Fig. 1). Resultant library preparation took an average of 45 min from purified genomic DNA, and 72 min from blood (Table 1). Thirdly, we developed much faster $2 \times 101$ cycle sequencing-by-synthesis on NovaSeq 6000 instruments (lllumina, average 11 h 12 min). This employed a custom instrument run recipe with maximally reduced cycle time, and SP flowcells, which were imaged only on one surface of each of two lanes. Fourthly, we developed a faster method for sequence alignment and variant calling (average 34 min for 120 GB of singleton genome sequence) that also had greatly improved analytic performance for SVs and CNVs (Dynamic Read Analysis for GENomics, DRAGEN v.3.7, Illumina)[16]. Finally, for generalizable, scalable clinical use, each of these components (sample accessioning, library preparation, library quality assessment, sequencing and variant calling) was integrated with a custom laboratory information management system and custom analysis pipeline (Enterprise Science Platform, L7 Informatics) that automated data transfers between steps.

The analytic performance and reproducibility of the combined method was evaluated in reference DNA samples in which benchmark variant sets have been established by the National Institute of Standards and Technology (NIST)[30,31]. The average time from DNA sample to completion of variant calling was 12 h and 42 min, 35% less than the previous minimum (Table 1)[16]. The analytic performance for single nucleotide variants (SNVs) and insertion-deletion oligonucleotide variants (indels) was also improved, with precision and recall values >99.4% (Table 2)[16].

The analytic performance of DRAGEN v.3.7 for structural variants (SVs, size >50 nt) and CNVs (size > 10 kb) was compared with the widely used methods Manta and CNVnator, respectively[32–34]. The latter require 2 h and 22 min longer cloud-based computation per sample than DRAGEN. The recall (sensitivity) of DRAGEN was considerably superior for insertion SVs (average 27% with Manta, 49% with DRAGEN) and deletion CNVs (average 9% with CNVnator, 88% with DRAGEN, Table 2). Since the NIST reference sample contains only 33 CNVs, the latter values should not yet be regarded as general estimates of analytic performance. However, chromosomal microarray, the most widely used diagnostic test for CNVs only detected one deletion CNV in this sample (Chr 7:142,824,207-142,893,380del, 3% sensitivity), which was classified as benign. It should also be noted that the software used to calculate analytic performance for

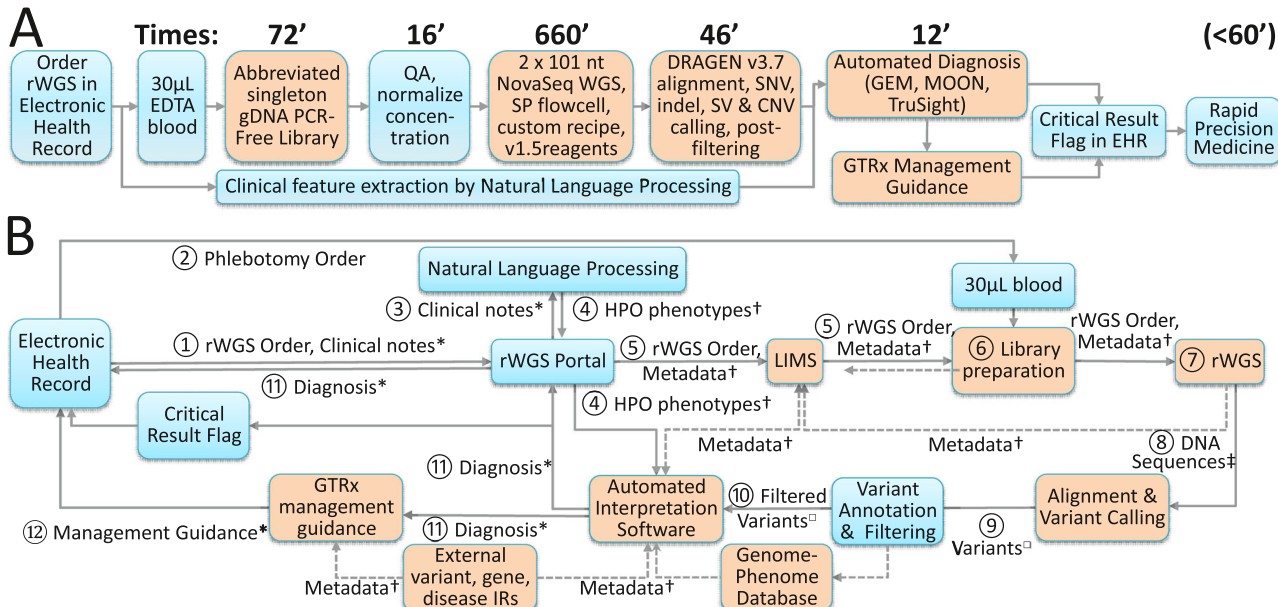

**Fig. 1 Flow diagrams of the technological components of a 13.5-hour system for automated diagnosis and virtual acute management guidance of genetic diseases by rWGS.** Innovations described herein are indicated by orange boxes. **A** The order and duration of laboratory steps and technologies. EHR Electronic Health Record, EDTA EthyleneDiamineTetraAcetic acid, gDNA genomic DeoxyriboNucleic Acid, PCR Polymerase Chain Reaction, QA Quality Assurance, nt Nucleotide, SNV Single Nucleotide Variant, indel insertion-deletion nucleotide variant, SV Structural Variant, CNV Copy Number Variant, GTRx Genome-to-Treatment. **B** Diagram of the information flow from order placement in the EHR to return of diagnostic results together with specific management guidance for that genetic disease. rWGS Portal: Custom software system for rWGS ordering, accessioning, chain-of-custody, and return of results (v.3.2). LIMS Custom laboratory information management system for rWGS, short tandem repeat profiling, confirmatory testing (Sanger sequencing and Multiplex Ligation-dependent Probe Amplification), and inventory management (L7 informatics). IR Information resource, *: HL7/FHIR or Continuity of Care Documents, †: JSON. ‡: bcl, □: vcf.

SV and CNV detection (Witty.Er), defines true positive matches more conservatively than in clinical diagnostic practice[35].

**Automated diagnosis of genetic diseases by genome sequencing.** Four further steps were needed for automated diagnosis of genetic diseases by WGS. Firstly, the patients' phenotypic features were automatically extracted from non-structured text fields in the electronic health record (EHR) using natural language processing (NLP, Clinithink Ltd.) through the date of enrollment for WGS[16]. The analytic performance of NLP and detailed manual review were compared with EHRs of ten children who received WGS. NLP identified an average of 89.8 Human Phenotype Ontology (HPO) features, including both exact matches and their hierarchical root terms (standard deviation (SD) 35.3, range 36–167; Table S1) per patient in ~20 s. Compared with manual review, which took several hours per record, the precision (positive predictive value, PPV) of NLP was 0.80 (SD 0.15, range 0.57–0.97) and recall (sensitivity) was 0.90 (SD 0.14, range 0.50–0.98). The performance of NLP in extraction of clinical features from EHRs and reasons for identification of false positive clinical features have been previously described[16].

Secondly, for each patient, the extracted HPO terms observed in the patient at time of enrollment were compared with the known HPO terms for all 7,103 genetic diseases with known causative loci[1]. Each genetic disease was assigned a likelihood of being the causative diagnosis based on the number of matching terms and their information content[16]. Thirdly, the pathogenicity of each variant detected by WGS was calculated by database lookup, if previously described, and by prediction of variant consequence for the associated protein[36–38]. Finally, a provisional genetic disease diagnosis was generated by rank ordering the integrated scores of phenotype similarity and diplotype pathogenicity. The provisional diagnosis contained none, one or a few genetic diseases. These four steps were integrated in three fully automated interpretation pipelines (InVitae MOON, Fabric GEM, and Illumina TruSight Software Suite, (TSS))[16,39].

We compared the diagnostic performance and reproducibility of this rWGS system, including the three interpretation pipelines, with blood samples from four affected children who had recently been diagnosed with a genetic disease by standard, clinical rWGS and manual interpretation (Table 1, S2). The automated systems correctly diagnosed the four infants. The average rank of the correct diagnosis was 1, 2 and 1 for MOON, GEM and TSS, respectively, and the ranges were 1–1, 1–4, and 1–1, respectively (Table S3). The mean number of candidate diagnoses returned were 16.5, 8 and 3.5 for MOON, GEM and TSS, respectively, and time to execution 10.3, 41.5 and 224.3 min, respectively (Table S3). The TSS time included DRAGEN 3.7 processing time, whereas the others did not. The average time from blood sample to provisional diagnosis result was 13 h 20.5 min, and fastest time was 13 h 13 min (Table 1). In each case, MOON had the fastest computation time.

**Development of an information resource for genetic diseases.** Manual interpretation is followed by writing a report of WGS results that includes information pertaining to the genetic diagnosis. This typically takes a genome analyst, genetic counselor, and laboratory director 1 or 2 h. Automated interpretation tools do not yet provide written reports. To make automated WGS more generalizable, we developed an information resource to automatically provide such information to front-line physician teams (Fig. 2).

First, we surveyed the numerous, existing web-based information resources for genetic diseases. Most were unstructured, incomplete, and not intended for use by front-line physicians. We obtained datasets from Online Mendelian Inheritance in Man (OMIM), Orphanet, Genetics Home Reference (GHR, now

**Table 1 Analytic performance, reproducibility, and duration of the major steps in automated diagnosis of genetic diseases by accelerated rWGS.**

| | 362 | 12878 | NA24385 | 930 | 1018 | AG928 | AG366 | AF414 | AI003 | AH638 Here | AH638 Std | CSD59F Here | CSD59F Std | CSD709 Here | CSD709 Std |
|---|---|---|---|---|---|---|---|---|---|---|---|---|---|---|---|
| **Sample / Run** | Ref.16 | 927 | 929 | 930 | 1018 | 1020 | 1204 | 1208 | 1218 | 1026 | 1027 | 477 | 480 | 478 | 479 |
| **Sample & Run Type** | DNA/Analytic performance | None | None | | | Blood/Retrospective | | | | Blood/Prospective | | | | | |
| **Diagnosis (Gene)** | None | None | None | | | *ALDOB* | *OTC* | *PCCA* | *SCN2A* | *SLC19A3* | | *MT-ATP6, SETD1A* | | *ADAMTSL2* | |
| **rWGS Methods** | Ref16 | Herein | Herein | | | Herein | | | | Here | Std | Here | Std | Here | Std |
| **SV & CNV ID Method** | None | ⚡ (DRAGEN) | MC / ⚡ | MC / ⚡ | MC / ⚡ | ⚡ | ⚡ | ⚡ | ⚡ | ⚡ | D3.5 | ⚡ | D3.5 | ⚡ | D3.5 |
| **Length of steps (min)** | | | | | | | | | | | | | | | |
| Sample Prep. Time | 151 | 50 | 45 | 41 | 50 | 74 | 71 | 69 | 67 | 80 | 1233 | 90 | 265 | 90 | 265 |
| Sequencing Time | 932 | 667 | 667 | 666 | 673 | 674 | 667 | 683 | 675 | 676 | 1067 | 687 | 1050 | 687 | 1050 |
| $1°/2°$ Analysis Time | 62 | 48 | 191 / 45 | 181 / 46 | 194 / 48 | 42 | 55 | 37 | 38 | 47 | 173 | 44 | 185 | 56 | 220 |
| Tertiary Analysis | n.a. | n.a. | n.a. | n.a. | n.a. | 10 | 14 | 13 | 13 | 10 | 87 | 12 | 126 | 21 | 131 |
| Total Time to Result | 1145 | 765 | 903 / 757 | 888 / 753 | 917 / 761 | 800 | 807 | 802 | 793 | 812 | 2560 | 833 | 1626 | 854 | 1666 |
| **Sequence metrics** | | | | | | | | | | | | | | | |
| Trimmed yield (Gigabases) | 149 | 192 | 178 | 186 | 189 | 165 | 176 | 80 | 135 | 187 | 162 | 182 | 144 | 174 | 153 |
| Reads with quality score >30 | 90.7% | 90.5% | 88.7% | 90.8% | 91.3% | 89.2% | 91.2% | 92.5% | 87.3% | 90.5% | 92.6% | 90.9% | 89.8% | 90.1% | 89.3% |
| Error rate | n.a. | 0.17% | 0.21% | 0.17% | 0.14% | 0.19% | 0.16% | 0.14% | 0.29% | 0.17% | 0.15% | 0.14% | 0.14% | 0.17% | 0.16% |
| Reads mapped | 98.9% | 96.7% | 96.8% | 96.8% | 97.2% | 96.0% | 96.9% | 89.0% | 94.8% | 96.2% | 99.1% | 96.1% | 99.1% | 95.5% | 98.6% |
| Duplicate reads | 8.5% | 11.6% | 10.8% | 12.9% | 13.9% | 15.2% | 15.5% | 23.2% | 14.5% | 13.7% | 11.4% | 15.8% | 10.4% | 14.9% | 13.6% |
| Mean insert size (Nt) | 345 | 395 | 438 | 449 | 445 | 440 | 426 | 496 | 468 | 465 | 423 | 491 | 467 | 502 | 460.5 |
| Average genome coverage | 47.5 | 52.3 | 49.1 | 49.9 | 50.5 | 44.5 | 47.44 | 19.46 | 36.7 | 49.1 | 45.7 | 46.9 | 40.7 | 45.1 | 41.5 |
| MIM genes w. >10X coverage of all coding domain Nt. | 95.8% | 97.6% | 96.4% | 96.7% | 97.1% | 94.9% | 95.8% | 4.2% | 92.2% | 90.1% | 95.5% | 94.6% | 94.5% | 94.7% | 94.6% |
| **Variant metrics** | | | | | | | | | | | | | | | |
| Nt Variants (1000 s) | 4733 | 4834 | 4838 | 4838 | 4837 | 4857 | 3789 | 3904 | 4851 | 4691 | 4690 | 4852 | 4852 | 4916 | 4910 |
| Variants passing Quality Metrics | 96.8% | 98.9% | 99.1% | 99.1% | 99.0% | 99.0% | 99.0% | 98.4% | 98.6% | 98.9% | 98.9% | 99.0% | 98.9% | 98.9% | 98.9% |
| Coding domain variants | 0.58% | 0.51% | 0.52% | 0.52% | 0.52% | 0.53% | 0.52% | 0.52% | 0.51% | 0.52% | 0.52% | 0.52% | 0.52% | 0.53% | 0.53% |
| Nt insertions & deletions | 17.5% | 19.7% | 19.7% | 19.7% | 19.6% | 19.5% | 19.6% | 18.9% | 19.4% | 19.6% | 19.6% | 19.7% | 19.7% | 19.7% | 19.7% |
| Transition/transversion ratio | 2.02 | 2.03 | 2.02 | 2.02 | 2.02 | 2.03 | 2.03 | 2.03 | 2.03 | 2.03 | 2.03 | 2.03 | 2.03 | 2.03 | 2.03 |

Analytic and diagnostic reproducibility were examined for sample 362 from 19.5-h rWGS (16), reference samples NA12878 and NA24385, four retrospective samples/diagnoses (AG928/Hereditary fructose intolerance (compound heterozygous, pathogenic (P) SNVs in aldolase B [*ALDOB* c.448 G > C, c.524 C > A]); AG366/Ornithine transcarbamylase deficiency (hemizygous, de novo, P, SNV in ornithine transcarbamylase [*OTC* c.275 G > A]); AF414/Propionic acidemia (homozygous, likely pathogenic (LP) indel in $\alpha$-subunit of propionyl-CoA carboxylase [*PCCA* c.1899 + 4_1899 + 7del]); AI003/Developmental and epileptic encephalopathy 11 (heterozygous, de novo, LP SNV in the $\alpha$2-subunit of the voltage-gated sodium channel [*SCN2A* c.4437 G > C]), and three prospective samples (AH638/Thiamine metabolism dysfunction syndrome 2 (homozygous, P, frame-shift variant in solute carrier 19, member 3 [*SLC19A3* c.597dup]); CSD59F (heterozygous, P, SNV in the mitochondrial ATP synthase 6 gene [*MT-ATP6* m.8993 T > C]), and CSD709/ Geleophysic dysplasia (compound heterozygous SNVs in ADAMTS-like 2 [*ADAMTSL2* c.338 G > T and c.1851C > A]), which received rWGS both with the 13.5-h method (Herein) and standard, singleton or trio, clinical rWGS (Std) (Table S2). Ref.16: Reference 16. Sample 12878: Sample NA12878. ID: Identification. Here: Herein. $1°/2°$ analysis time: Conversion of raw data from base call to FASTQ format, read alignment to the reference genomes and variant calling. Tertiary analysis: Time of automated interpretation to provisional diagnosis (most rapid of three systems run in parallel (MOON, Illumina TruSight Software Suite and GEM). SV and CNV detection methods: MC: Manta and CNVnator). ⚡: DRAGEN version 3.7. D3.5: DRAGEN version 3.5.3. MIM: Mendelian inheritance in man. Nt: Nucleotide. Gene symbols are shown in italics. Variant section headers are shown in bold.

**Table 2 Comparison of the analytic performance of standard, clinical rWGS, and the 13.5-h method.**

| Variant type | Performance metric | NA12878 | | | NA24385 | | |
|---|---|---|---|---|---|---|---|
| | | Variant number | v.2.5 | 🐆 | Variant number | MC | 🐆 |
| SNV | Precision | 3,258,654 | 99.8% | 99.9% | 3,440,606 | n.a. | 99.7% |
| | Recall | | 99.7% | 99.9% | | n.a. | 99.3% |
| indel | Precision | 490,488 | 99.0% | 99.6% | 553,766 | n.a. | 99.4% |
| | Recall | | 95.5% | 99.4% | | n.a. | 98.6% |
| SV deletion | Precision | n.a. | n.a. | n.a. | 4203 | 91.7% | 97.1% |
| | Recall | | n.a. | n.a. | | 57.3% | 61.7% |
| SV insertion | Precision | n.a. | n.a. | n.a. | 5444 | 99.0% | 98.4% |
| | Recall | | n.a. | n.a. | | 27.4% | 49.3% |
| CNV deletion | Precision | n.a. | n.a. | n.a. | 33 | 83.3% | 100.0% |
| | Recall | | n.a. | n.a. | | 9.1% | 87.9% |

The analytic performance of DRAGEN v.3.7 (🐆) for SNVs and indels was compared with DRAGEN v2.5, the prior method, in reference samples NA12878 and NA24385, using NIST benchmark genotypes[16]. The analytic performance of DRAGEN v3.7 for SVs and CNVs was compared with Manta and CNVnator (MC) in triplicate libraries in reference sample NA24385, using NIST benchmark genotypes. SV and CNV evaluations used Witty.Er, with default settings except event reporting [–em cts])[35]. SVs were of size >50 nt and CNVs >10 kb.

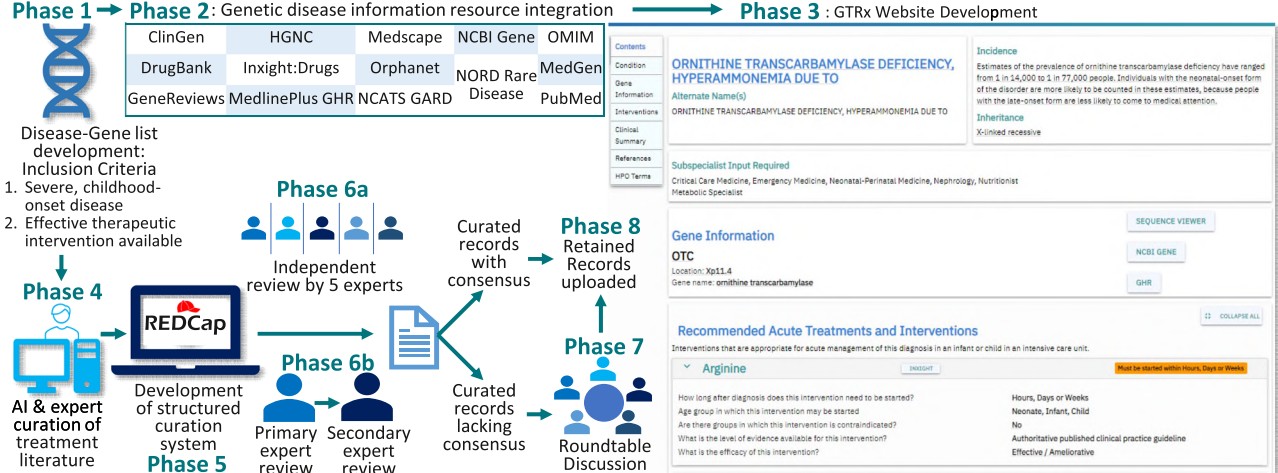

**Fig. 2 Flowchart of the development of Genome-To-Treatment (GTRx), a virtual system for acute management guidance for rare genetic diseases.** Phase 1 - Compilation of a comprehensive gene-genetic disease list for severe, childhood-onset conditions in which an established treatment was available. Phase 2, integration of 13 information resources pertaining to rare genetic diseases. Phase 3, development of the GTRx web resource containing the integrated information resources. Phase 4, automated, artificial intelligence (AI)-based searching and manual curation of published evidence of treatments for each condition by three companies. Phase 5, development of a custom REDCap system for structured assessment of genes, disorders, and therapeutic interventions. Phase 6a, independent manual review of curated interventions and assertions for the first 15 pilot gene-disease pairs by five experts. Phase 6b, primary and secondary reviews of the remaining gene-disease pairs. Phase 7, round-table discussion of records lacking consensus. Phase 8, upload of retained consensus records to the GTRx web resource.

MedLinePlus), DrugBank v5.0, the National Center for Advancing Translational Sciences resources (Inxight:Drugs, Genetic and Rare Disease Information Center (GARD), Medscape, NORD's Rare Disease Database, the National Center for BI resources (Gene, ClinVar, ClinicalTrials.gov, GeneReviews, and MedGen), the Cochrane Database of Systematic Reviews, and PubMed[40–52]. We built transformation pipelines with the Konstanz Information Miner (KNIME) to match entries, normalize, and merge them[53]. Unifying gene definitions were from RefSeq, and genetic disease definitions from mappings between OMIM and Orphanet[40,41,54]. OMIM identities were used except where there was only an Orphanet entry. Unifying HPO phenotypes were mapped to OMIM, Orphanet and GARD[40,41,55]. We developed a web resource, Genome-To-Treatment (GTRx, http://gtrx.rbsapp.net/) to automatically display this information and link it to automated WGS results on a gene-by-gene basis (Figs. 2 and S1–S3).

**Development of an electronic acute management support system.** Clinical implementation of rWGS has shown that rapid molecular diagnosis alone may be insufficient to improve outcomes in diseases with effective treatments that progress rapidly to severe morbidity or mortality if untreated[10]. Front-line physicians are often unfamiliar with treatments for rare genetic diseases. Sub-specialist or multi-disciplinary consultation may materially delay treatment. We therefore developed a virtual acute management guidance system for rare genetic diseases with effective treatments, the Treatabolome, that was integrated into the information resource described above (Fig. 2)[56].

For common diseases, it would have been relatively straightforward to integrate DrugBank Plus, Food and Drug Administration (FDA) indications, and additional resources such as InXight Drugs and ClinicalTrials.gov. However, most drug treatments for rare childhood genetic diseases are prescribed off-label[57]. Furthermore, specialized diets, dietary supplements, and surgeries, which are not subject to FDA review, are also critical components of treatment for rare childhood genetic diseases. Devices are another important class of intervention for children in ICUs. While devices are subject to FDA review,

approvals are not tied to genetic disease diagnoses. We reviewed publicly available information resources for rare childhood genetic disease interventions, including published clinical practice guidelines, OMIM, Orphanet, GHR, GARD, PubMed, GeneReviews, American College of Medical Genetics (ACMG) Newborn Screening ACTion (ACT) sheets, Acute Illness Materials developed by the New England Consortium of Metabolic Programs, and ActX[40–60] We discovered a lack of broadly applicable instruments to measure rare genetic disease progression or outcomes, or orphan treatment effects, such as quality of life or real-world outcomes[61]. Many genetic diseases lacked sufficient ground truth knowledge of variability in natural history if untreated, or relative effectiveness of standard of care treatments. Evidence of efficacy was generally short-term and from single-arm case reports or small case series. There was no consensus scheme for classification of the efficacy of treatments nor the quality of the evidence supporting efficacy. The best existing resource for treatment guidance for many different types of genetic diseases was GeneReviews[43]. However, it was unstructured and subject to many of these limitations. Content variability was compounded by review of each disease by a different set of experts. It did not review all childhood genetic diseases with effective treatments, and chapters were revised only every several years. It was necessary, therefore, to create a structured database of rare childhood genetic disease interventions that complied with the Findable, Accessible, Interoperable and Reusable (FAIR) guiding principles de novo[56,62].

In light of substantial shortcomings of normalized knowledge of genetic disease treatments, we defined the narrowest scope for an electronic acute disease management support system (Fig. 2): It was intended to guide initial, optimal treatment for critically ill children in ICUs at time of genetic disease diagnosis by rWGS. It was limited to diseases with effective treatments and rapid progression in the absence of those treatments. It was designed for use by front-line intensivists, neonatologists and hospitalists during the time interval between return of rWGS results and provision of authoritative subspecialist guidance or transfer to a tertiary or quaternary hospital. We assumed that front-line physicians were unlikely to have treated a child with that disease in that setting before. We also assumed that they would have limited genomic literacy, lack of familiarity with existing genetic disease information resources, and insufficient time to synthesize treatments by literature perusal. While limited in scope, we sought interoperability with broader future use.

Second, we identified 358 genes associated with 563 genetic diseases, representing 8% of 7103 single locus genetic diseases, that met the following criteria: acute, childhood presentations that were likely to lead to neonatal, pediatric or cardiovascular ICU admission; having somewhat effective treatments; high likelihood of rapid progression without treatment; and, diagnosable by rWGS (Figs. 2, 3, Supplementary Data 1). They were identified by a survey of our clinical rWGS experience in ~3500 cases, and from expanded newborn screening lists developed by several groups[2–7,10–16,18,28,63–67].

Third, we determined the minimal data elements needed by front-line physicians upon receipt of an rWGS result. In the setting of a newly diagnosed genetic disease in a critically ill child, they needed to know the indicated interventions, optimal time to administration, efficacy, evidence for efficacy, contraindications, and natural history without treatment (Box 1)[56]. We assumed that adequate resources existed to provide guidance about drug dosing, frequency, route of administration, drug-drug interactions or labeled contraindications.

Fourth, we required that the virtual, acute disease management guidance system (GTRx) was authoritative and consensus-driven. For each genetic disease, we indexed the full text of all MEDLINE/PubMed references that mentioned a drug, device, diet or surgery used to treat the disease using three artificial intelligence based search engines (Mastermind, Genomenon; Rancho Biosciences, Epam Systems, Fig. 2)[51]. The resultant datasets were manually curated for relevance and specificity, and to extract the required data elements (Supplementary Data 2). The manually curated datasets and links to the information resource were integrated into a custom Research Electronic Data Capture (REDCap) survey for expert review (Figs. 2 and S4–S7, Supplementary Data 2)[68]. Each disease and intervention were reviewed by a panel of five highly experienced, pediatric biochemical geneticists to answer seven categorical questions (Figs. 2 and S4–S7, Box 1, Supplementary Data 2). The first 15 genetic diseases and 200 associated interventions were independently reviewed by each expert. 52.8% of intervention reviews were concordant. Discordant responses were discussed virtually by the moderated panel (Table S4). After discussion, the panel agreed upon 189 (99%) of the first 190 (Fig. 2, Table S4), and retained 84 interventions. There were three reasons for rejection of the remaining 106 nominated interventions: inadequate evidence for efficacy (25%, 27), incorrect treatment for that disorder (27%, 23), and insufficient specificity to warrant inclusion (19%, 20). Reviewers also examined the age category in which each intervention was suitable (neonate, infant, child), optimal time after diagnosis for initiation (hours, days/weeks, years), significant contraindications in subgroups of patients, efficacy of the intervention in that disease (curative, effective/ameliorative, still in trials/unproven), and level of published evidence for each intervention (authoritative clinical practice guideline, cohort study(ies), case report(s)). Consensus was reached for each question for each retained intervention. In addition, the experts identified appropriate consulting sub-specialists for each condition and emergency treatment notification flags, if any, that should accompany diagnostic reports (Figs. S1–S3).

Informed by experience with the first 15 disease genes, a total of 563 disorder-gene dyads underwent single primary, and secondary reviews by members of the same panel (Fig. 2). Primary reviews required 1–5 h of effort by an expert medical geneticist, and secondary reviews required 1 h of effort. Interventions lacking consensus were discussed by the five reviewers. Consensus was required for retention (Supplementary Data 3). For disorders that reviewers or the moderator considered to require further input a final moderated review was performed by one or more pediatric subspecialists familiar with that disorder (Fig. 2). Examples of the latter included Timothy syndrome (cardiac electrophysiologist) and developmental epileptic encephalopathies (neonatal epileptologist). Review of 8,889 interventions and >5000 publications by the expert panel led to retention of 421 (75%) disorders and 1527 interventions (Fig. 3A, Supplementary Data 3), of which 118 (7.8%) were surgeries, 109 (7.2%) were diets or dietary supplements, 1046 (68.8%) were medications, 20 (1.3%) were devices, and 233 (14.8%) were of other types (Fig. 3A, Supplementary Data 3). 75 (5.0%) retained interventions were considered curative, and 1363 (90.6%) effective or ameliorative (Fig. 3A, Supplementary Data 3). Surgeries had the highest proportion of curative interventions (37.6%). The disease genes mapped to many organ systems and pathologic mechanisms (Fig. 3B).

The retained interventions and qualifying statements were incorporated into the GTRx information resource as a prototypic acute management guidance system for genetic diseases that meets FAIR principles[56,62] (Figs. 2 and 3, https://gtrx.radygenomiclab.com).

**Physician perception of the utility of GTRx.** The clinical utility, ease of use and ease of comprehension of the GTRx information resource and management guidance was evaluated by nine senior neonatologists and pediatric intensivists who were not involved in its design or development. On a 10-point Likert scale, their median perception as to whether they would use GTRx was 9, ease of use

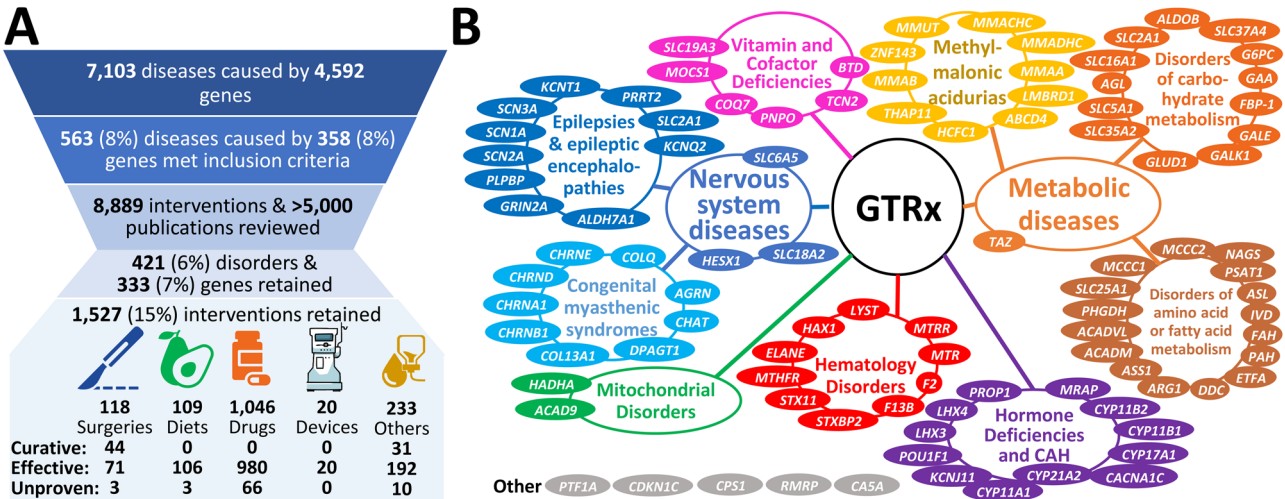

**Fig. 3 GTRx disease, gene, and literature filtering, and final content. A** A modified PRISMA flowchart showing filtering steps and summarizing results of review of 563 unique disease-gene dyads herein[86]. **B** Genetic disease types and disease genes featured in the first 100 GTRx genes reviewed herein.

**Box 1 | Minimal, structured data elements required for FAIR-compliant systematic literary reviews to create a virtual acute management support system for clinicians**

- Disease, gene, incidence, inheritance mode(s)
- Appropriate subspecialist consultant(s)
- Clinical summary / natural history of disease
- Set of appropriate acute treatments:
  - Drug(s)
  - Device(s)
  - Diet(s)
  - Surgical intervention(s)
- For each treatment:
  - Efficacy in this disease
    - Curative
    - Effective / Ameliorative
    - Still in Trials
    - Contraindicated
  - Evidence supporting efficacy in this disease
    - Authoritative published guidelines
    - Cohort study or studies
    - Case reports
  - Optimal timeframe to initiate after disease diagnosis
    - Hours
    - Days / Weeks
    - Years
  - Appropriate age group(s) in this disease
    - Neonates
    - Infants
    - Children
  - Contraindicated groups in this disease
  - Banner warning (if any)

was 9, and the utility of the information was 6 (Table S6). GTRx was perceived to meet clinical needs somewhat well. In response to specific feedback, the GTRx website was modified to increase ease of use, clarity, and to elicit ongoing feedback.

**Performance of the system for automated provisional diagnosis and virtual acute management support.** In four retrospective cases, the automated pipeline and electronic acute management support system identified the correct diagnosis in

13:13–13:27 h (Table 1). An independent physician evaluated the accuracy of the treatment guidance from the virtual acute management support system. In each case, the interventions were assessed to be correct and complete (Table 1, Table S1).

We prospectively compared the performance of the 13.5-h system for automated provisional diagnosis and the GTRx electronic acute management support system with the fastest standard clinical methods in three infants (Table 1, Fig. 4). The first prospective case, AH638, was a 6-week-old male admitted to the neonatal ICU with extreme irritability and inconsolable crying. Brain magnetic resonance imaging revealed widespread, symmetric hypodense lesions[69]. Electroencephalography (EEG) revealed frequent seizures. The proband's elder sister died nine years earlier, at 11 months of age, after presenting at the same age with the same symptoms and findings. WGS was not available at that time, and she died of progressive developmental epileptic encephalopathy without an etiologic diagnosis. His parents were first cousins. The prototypic methods provided a provisional diagnosis in 13 h and 32 min. The diagnosis was autosomal recessive thiamine metabolism dysfunction syndrome 2, biotin- or thiamine-responsive type (Online Mendelian Inheritance in Man (MIM) #607483, https://www.omim.org/entry/607483) associated with a pathogenic, homozygous, frameshift variant in the thiamine transporter 2 gene (*SLC19A3* c.597dup, p.His200fs, https://www.ncbi.nlm.nih.gov/clinvar/variation/533549/?oq=SLC19A3[gene]+AND+c.597dupT[varname]+&m=NM_025243.4(SLC19A3):c.597dup%20(p.His200fs)). The provisional diagnosis was immediately communicated to the neonatologist of record. Effective treatments (biotin and thiamine supplements) were initiated within 3 h of diagnosis[69]. He responded to treatment and was alert, tranquil, and bottle feeding within 6 h of treatment. Standard clinical rWGS methods recapitulated the diagnosis in 42 h and 39 min. He had no further seizures and was discharged home after 3 days. At fifteen months of age, he has had no further seizures. He is making developmental progress but has delayed motor and language development.

The second patient, CSD59F, a male, was admitted to the neonatal ICU on day of life 6 after his mother noticed abnormal, jerking movements (Table 1, Fig. 4). EEG disclosed frequent seizures. He had hypocalcemia (6.1 mg/dL, reference range 7.6–10.4 mg/dL) and hyperphosphatemia (11.2 mg/dL, reference range 4.3–9.3 mg/dL). The prototypic methods yielded a provisional diagnosis of Leigh syndrome (MIM#256000, https://www.omim.org/entry/256000) in 15 h and 5 min. Peripheral blood

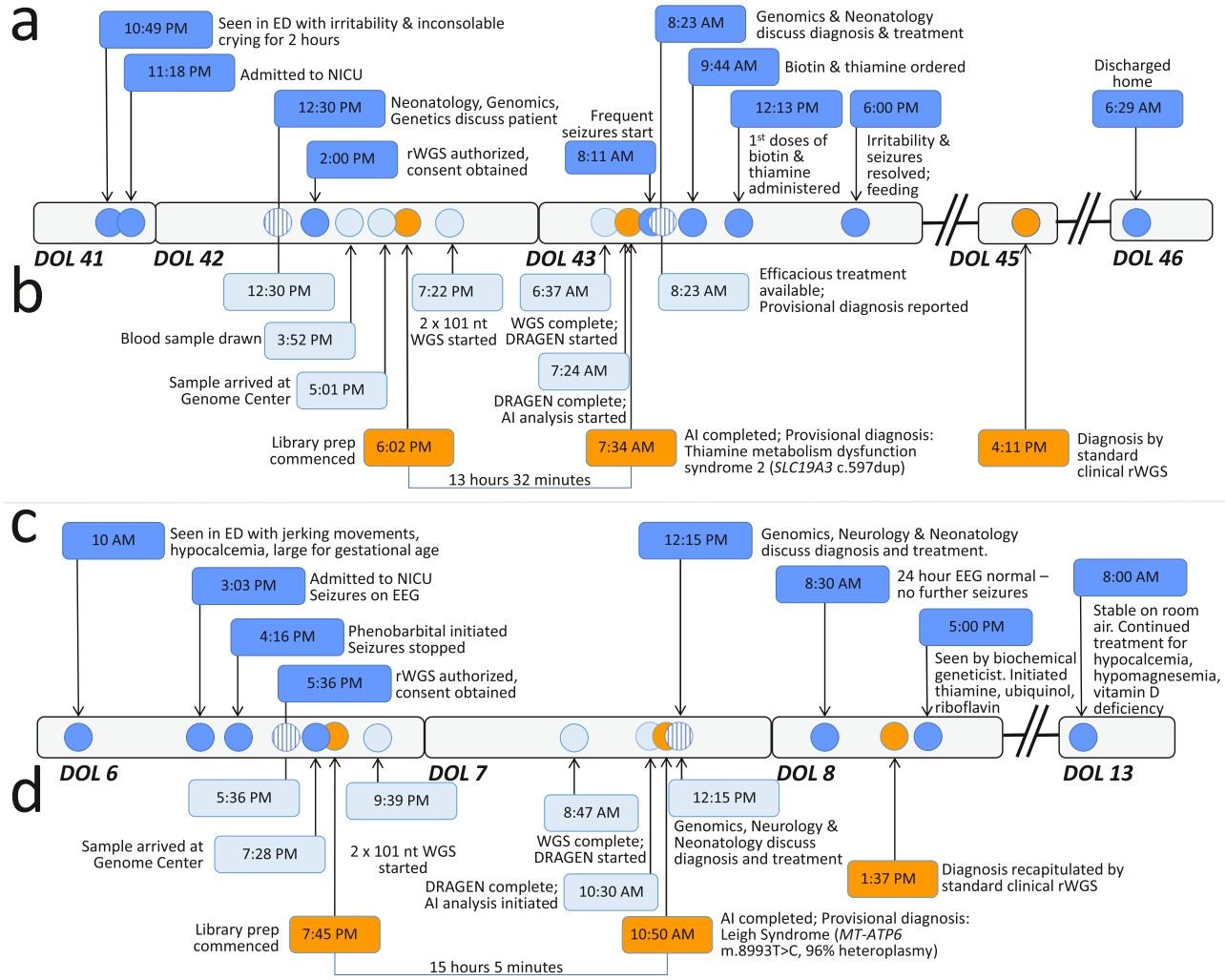

**Fig. 4 Clinical course and diagnostic timeline of two critically ill infants who received 13.5-h rWGS and confirmatory standard diagnostic rWGS.** Clinical (**a** and **c**, dark blue circles) and diagnostic timelines (**b** and **d**, light blue circles) of infants AH638 (**a**, **b**) and CSD59F (**c**, **d**), who received both standard, clinical rWGS and the 13.5-h methods. ED Emergency Department, EEG Electroencephalogram, AI Artificial intelligence, DOL Day of life. Circles with vertical lines indicate interactions between neonatology, genomics, and biochemical genetics.

DNA had de novo 96% heteroplasmy (1351/1402 reads) for a well-established, pathogenic variant in the mitochondrial ATP synthase subunit 6 gene (*MT-ATP6* m.8993 T > C, p.Leu156Pro, https://www.ncbi.nlm.nih.gov/clinvar/variation/9642/?oq=MT-ATP6[gene]+AND + m.8993 T%3EC[varname] +&m=NC_012920.1:m.8993 T%3EC). Leigh syndrome is associated with infantile seizures[70]. The provisional diagnosis of Leigh syndrome was immediately communicated to the neonatologist of record. A heterozygous variant of uncertain significance was also identified in the SET domain-containing protein 1 A gene (*SETD1A* c.4105 G > A, p.Gly1369Arg, https://www.ncbi.nlm.nih.gov/clinvar/variation/834092/?oq=SETD1A[gene]+AND + c.4105 G%3EA[varname]+&m=NM_014712.3(SETD1A):c.4105 G%3EA%20(p.Gly1369Arg)). Pathogenic variation in *SETD1A* is associated with autosomal dominant, Early-Onset Epilepsy with or without developmental delay (MIM #618832, https://www.omim.org/entry/618832). This finding was not reported provisionally. Standard clinical rWGS methods recapitulated these findings in 42 h and 5 min, and a final report was issued of both findings. Seizures remitted with phenobarbital. He was seen by a subspecialist in mitochondrial diseases within 48 h of admission, and initiated on thiamine, ubiquinol and riboflavin

supplementation. He was discharged in stable condition with no further seizures on day of life 23.

The third patient, CSD709, a male, was admitted to the neonatal ICU on the first day of life with respiratory failure, lactic acidosis, encephalopathy, hypotonia, multiple congenital anomalies (short long bones in the upper and lower limbs, posteriorly rotated ears, dysmorphic knees, and congenital heart disease (pulmonary artery stenosis, pulmonary arterial hypertension, aortic valve stenosis, and right ventricular hypertrophy))(Table 1). rWGS was completed in 14 h and 14 min by the prototypic methods but did not yield a provisional diagnosis. Standard clinical rWGS methods completed in 27 h and 46 min. Both disclosed a heterozygous, likely pathogenic, SNV in a disintegrin and metalloproteinase with thrombospondin motifs-like protein 2 (*ADAMTSL2* c.338 G > T, p.Arg113Leu, https://www.ncbi.nlm.nih.gov/clinvar/variation/1326072/?oq=ADAMTSL2[gene]+AND + c.338 G%3ET[varname]+&m=NM_014694.4(ADAMTSL2):c.338 G%3ET%20(p.Arg113Leu)) that had previously been reported in patients with geleophysic dysplasia (MIM# 231050, https://www.omim.org/entry/231050?search=231050&highlight=231050) as a compound heterozygous or homozygous change[71,72]. The variant call file (vcf) did not contain a second variant in *ADAMTSL2*. However, *ADAMTSL2* is located in a region that is

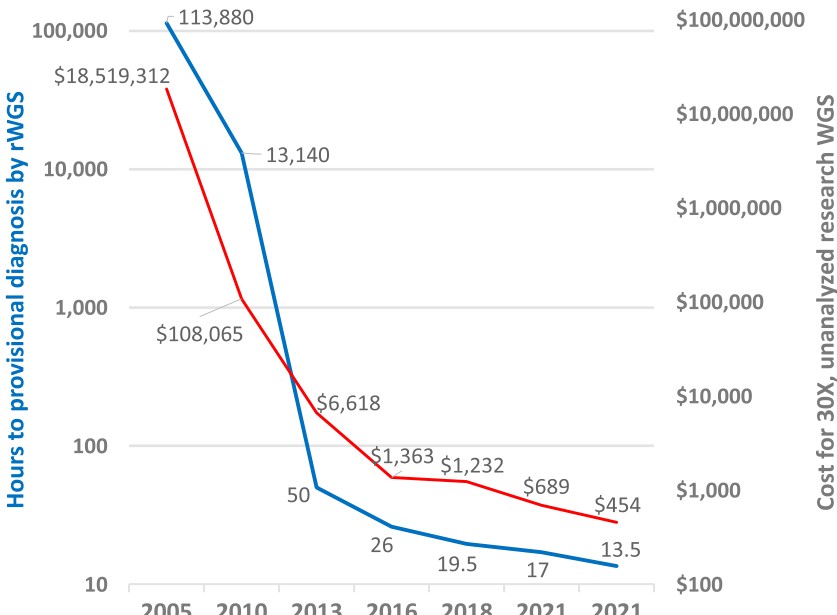

**Fig. 5 Decreasing cost of research WGS (red line) and time to provisional diagnosis of rapid, clinical WGS (blue line) of WGS, 2005–2021.** [13, 15-17] (https://www.genome.gov/about-genomics/fact-sheets/Sequencing-Human-Genome-cost). Source data are provided as a Source Data file.

affected by segmental duplication. Manual inspection of aligned *ADAMTSL2* reads revealed a second heterozygous, likely pathogenic variant (c.1851C > A, p.Cys617Ter, https://www.ncbi.nlm.nih.gov/clinvar/variation/1326007/?oq=ADAMTSL2 [gene]+AND + c.1851C%3EA[varname]+&m=NM_014694.4 (ADAMTSL2):c.1851C%3EA%20(p.Cys617Ter)). Both variants were confirmed to be in trans by orthogonal methods and a diagnosis of geleophysic dysplasia was reported after 14 days.

## Discussion

The cost and turnaround time of WGS have decreased dramatically since its advent 15 years ago (Fig. 5). The first human genome took 13 years to complete[73]. Here, we described and examined the performance of a 13.5-h, autonomous system for genetic disease diagnosis by rapid WGS and virtual, specific management guidance. This is the fifth reduction in the minimal time to diagnosis by WGS since 2012[13,15–17]. While this manuscript was under review, a 7-h, method for genetic disease diagnosis by long-read WGS was published[74]. The rationale for continuing to pursue faster diagnosis was strikingly exemplified in the first infant to receive 13.5-h WGS[69]. He was diagnosed in 13 h and 32 min with a disorder that is both treatable and extremely rapidly progressive. Had his diagnosis been delayed until the standard rWGS result (42.5 h) he would likely have had significant, permanent neurologic damage. In contrast, his sister died without an etiologic diagnosis, and thus, without effective treatment. The experience in this family was not unique[10]. Since it is not possible to determine a priori which cases require such rapidity, the general practice has been to provide the fastest turnaround possible for all critically ill infants and children or those with rapid clinical progression in ICUs and who have diseases of unknown etiology[3–5,7] (https://www.blueshieldca.com/bsca/bsc/public/common/PortalComponents/provider/StreamDocumentServlet?fileName=PRV_WholeExome_Sequen.pdf)[22]. At current volume of ~100 cases per month, our median turnaround time for critical cases is 30–36 h. In clinical production in three cases, we have found that these methods have reduced this by a factor of two.

There is now strong evidence that diagnosis of genetic diseases by rWGS improves outcomes of infants and children in regional ICUs, irrespective of presentation or health system[2–8,10–18,28,69,74–76]. As a result, diagnostic rWGS is being implemented for such children in England, Wales, and Germany, by Anthem/BlueCross/BlueShield in the USA, and by Medicaid in California and Michigan[18] (https://www.genomicsengland.co.uk/matt-hancock-announces-5-million-genomes-within-five-years/) (https://www.blueshieldca.com/bsca/bsc/public/common/PortalComponents/provider/StreamDocumentServlet?fileName=PRV_WholeExome_Sequen.pdf) (https://dejure.org/gesetze/SGB_V/64e.html) (https://files.medi-cal.ca.gov/pubsdoco/bulletins/artfull/ips202112.aspx) (https://www.michigan.gov/documents/mdhhs/MSA_21-33_732848_7.pdf). Scalability of rWGS in routine practice is, therefore, as important as turnaround time. The 13.5-h system for genetic disease diagnosis incorporated several innovations that enhance scalability and reproducibility. These included automated interpretation, which is extremely important since there are insufficient molecular pathologists, molecular laboratory directors, genetic counselors and clinical genome analysts for manual interpretation of results from all of the children for whom rWGS is being implemented[16]. As sequencing costs decrease (Fig. 5), manual interpretation and reporting are becoming the largest component of the expense of diagnostic rWGS[33]. Herein, we compared three, cloud-based methods for autonomous genetic disease diagnosis, providing the opportunity for cross checking of results. The only requirements for implementation of this system are an EHR, internet access, and a regional diagnostic lab with a suitable sequencer. In the future we envisage cloud-based, automated interpretation that is supervised by a laboratory director and supplemented with centralized, manual interpretation for edge cases[16]. We recently evaluated the diagnostic performance the automated interpretation system GEM, in 193 children with suspected genetic diseases[33]. In 92% of cases, GEM ranked the correct gene and variant in the top two calls, including structural variant diagnoses. However, to date the full 13.5-h system has been evaluated only in four retrospective and six prospective cases. Further studies are needed for clinical validation, such as reproducibility, performance with all patterns of inheritance, examination of the relative diagnostic performance of automated

methods compared with traditional manual interpretation, and to understand the proportion of edge cases.

Another innovation of the system described herein was ability to diagnose genetic diseases associated with most major classes of genomic variants. Hitherto, diagnostic speed was achieved at the expense of limitation to small (nucleotide) variants, which represent 75–80% of genetic disease diagnoses[14] (https://www.blueshieldca. com/bsca/bsc/public/common/PortalComponents/provider/Stream DocumentServlet?fileName=PRV_WholeExome_Sequen.pdf)[22]. Here, we used methods for library preparation, variant calling, and automated interpretation that enabled structural and copy number variant (SV, CNV) diagnoses with improved performance[39]. It should be noted, however, that recall (sensitivity) for SVs and CNVs remain a weakness of short read sequencing (range 49–88%). The consequences of this for genetic disease diagnosis is not yet known. Further studies are needed to compare the diagnostic performance of these methods versus hybrid methods with short read sequencing and complementary technologies, such as long-read sequencing and optical mapping[74,77,78].

Finally, the 13.5-h system featured a virtual clinical decision support system, Gene-to-Treatment (GTRx[SM], https://gtrx. radygenomiclab.com) to decrease variability or delayed implementation of specific treatment following diagnosis of rare genetic conditions[57]. Hitherto, use of rWGS has been almost entirely in ICUs in regional, academic, tertiary, or quaternary centers with specialist neonatologists and access to a full range of subspecialist consultants. Lack of familiarity with management of specific, rare genetic diseases leads to delays in consultation and missed opportunities for treatment that defeat the goal of rapid diagnosis. GTRx was developed both to increase the proportion of children who receive optimal, immediate treatment and to facilitate broader use of rWGS, such as in local birthing hospitals staffed by front-line neonatologists. In California, for example, while 18% of newborns are admitted to level II and III NICUs in community birthing hospitals, only 2% of newborns are transferred to regional, level IV neonatal intensive care units. Transfers are often delayed since there is a strong desire to provide care for the newborn at the same location as his or her mother, and it is often not readily apparent that subspecialist care is required. In many regions of the US, geographic isolation limits transfer. GTRx adheres to the technical standards developed by the ACMG for diagnostic genomic sequencing[36–38,79]. The most recent guidelines suggest the addition of references to treatments in reports of genes associated with a treatable genetic disorder[79].

The extent to which rare genetic diseases did not have organized management guidance was surprising. For many, the mechanism of disease remained unclear, and the treatment literature comprised only case reports or small case series. Most interventions were off label. Furthermore, no general schema existed whereby to classify the relative efficacy of interventions for specific genetic disorders nor the quality of the evidence for efficacy. We developed methods to extract and transform treatment data from the literature. We developed a categorical framework for nomenclature, efficacy, evidence, indicated population, immediacy of initiation of treatment and warnings. We used tiered reviews, facilitated by artificial intelligence and REDCap, and expert consensus to retain efficacious interventions. The resultant prototypic acute management guidance tool and information resource, GTRx, was intended for use by front-line neonatologists and intensivists upon receipt of results of rWGS for children under their care in ICUs. It did not require genomic or genetic literacy. Version 1 of GTRx covers 457 genetic disorders that cause infant or early childhood ICU admission and that have somewhat effective, time-delimited treatments. GTRx is publicly available for research use at present. We plan to examine and refine the clinical utility of GTRx through research studies of rWGS in a variety of healthcare settings to qualify it for clinical use.

Version 1 of GTRx does not cover all genetic diseases of known molecular cause, that can be diagnosed by rWGS, can lead to ICU admission in infancy, and have effective treatments. In addition, the literature related to disease treatments is continually being augmented. While pediatric geneticists were optimal sub-specialists for initial review of disorders and interventions, many would benefit from additional sub- and super-specialist review. We plan to address these limitations in future versions of GTRx, with ongoing, expert, open, community-based review. In addition, recent evidence supports the use of rWGS for genetic disease diagnosis and management guidance in older children in pediatric ICUs[5]. It is desirable to include these conditions in future versions. There are several, additional, complementary information resources that would enrich GTRx, such as ClinGen, the Genetic Test Registry, and Rx-Genes[80,81] (https://www.ncbi.nlm. nih.gov/gtr/). Finally, there are many clinical trials of new interventions for infant-onset, severe genetic disorders, particularly genetic therapies. For disorders without a current effective treatment, it is desirable to include links to enrollment contacts for those clinical trials.

Currently, pathogenicity guidelines help molecular laboratory directors standardize how many and which genome findings to report. GTRx will help standardize the reporting of variants of uncertain significance (VUS), which, at present, is predicated on the goodness of fit of the patient's presentation and the phenotype associated with the variant containing gene. In the setting of GTRx, VUS reporting will be further prioritized by the availability of an effective treatment for the associated disease, akin to variant tiering in oncology[82]. The GTRx information resource will simplify the writing of rWGS reports, extending the ability to automate diagnosis. Thus, for each automated WGS result, GTRx provides access to information about each genetic disease, including inheritance, incidence, symptoms and signs, progression, complications and outcomes, and the causal gene, including function, and mechanism of disease.

As genomic literacy and experience evolves, physicians increasingly wish to reinterpret findings themselves, dynamically adjusting the scope of review on a case-by-case basis[83]. In the longer term, automated genome interpretation and virtual management guidance have the potential to empower dynamic physician re-analysis[84]. In the future, we envisage GTRx to evolve into a virtual physician assistant, equipping physicians to dynamically explore the goodness of fit of observed and various candidate disease phenotype sets[16]. Where associated diplotypes are incomplete or include variants of uncertain significance, GTRx will allow ordering of confirmatory tests(https://www.ncbi.nlm. nih.gov/gtr/). GTRx will also assist physicians in decision making with regard to a possible trial of treatment for a potential diagnosis, guided by the risk: benefit ratio. This is particularly important for critically ill patients where a genetic etiology is strongly suspected but genome findings are insufficient for strict molecular diagnosis. GTRx will also assist front-line physicians to communicate with families about the ramifications of rare genetic disease diagnoses. GTRx is part of a major trend in medicine— adding artificial intelligence to physician competency to deliver "high-performance medicine"[85].

In summary, we describe a 13.5-h prototypic system for automated genetic disease diagnosis and acute management guidance. The system was designed to expand the use of rWGS by front-line physicians caring for critically ill infants and children in ICUs. At present, the system is prototypic and encompasses only ~500 genetic diseases that progress rapidly, and for which effective treatments are available. Upon validation of clinical utility, we envisage expansion of the system to all genetic diseases

and to dynamic filtering, enabling front-line physicians to play a much more active role in evaluating potential genetic etiologies and their consequent therapies in their patients.

## Methods

**Study design.** This study reports results from human subject research approved by the institutional review board at Rady Children's Hospital, San Diego, and the University of California–San Diego, which were performed in accordance with the Declaration of Helsinki. Informed, written consent was obtained from at least one parent or guardian of the participating infants, including permission to disclose indirect identifiers. Families were not compensated for participation. Datasets were obtained from four retrospectively studied infants (age less than one year, two male and two female) and three prospectively studied male neonates (aged less than 28 days) to test the analytic, diagnostic, and clinical management performance of the 13.5-h method. Ten cases (six male and four female, seven neonates, two older infants, and one 14-year old) used to verify the analytic performance of the clinical natural language processing were identified from research study populations[16]. Four retrospective cases were identified from recent clinical operations at Rady Children's Institute for Genomic Medicine (RCIGM). All had received recent diagnoses by rWGS, performed in the RCIGM CLIA/CAP laboratory, and blood sample retains were used for comparative re-analysis by the 13.5-h method. Three prospective cases were also ascertained from RCIGM clinical operations. Prospective cases received both standard rWGS performed according to CLIA/CAP standards and the prototypic 13.5-h method concomitantly. Provisional results from the prototypic 13.5-h method were returned to the attending neonatologist before confirmation by the standard method in accordance with a determination of "nonsignificant risk" by the FDA in response to an Investigational Device Exemption pre-submission enquiry for the antecedent study in April 2014 (https://www.illumina.com/content/dam/illumina-marketing/documents/products/appnotes/illumina-dna-pcr-free-wgs-app-note-770-2020-006.pdf). This study also reports results of a quality improvement project for diagnostic rWGS performed at Rady Children's Institute for Genomic Medicine (RCIGM) laboratory in conformity with the College of American Pathologists (CAP) and Clinical Laboratory Improvement Amendments (CLIA) standards.

**Natural language processing and phenotype extraction.** Human phenotype ontology (HPO, https://github.com/obophenotype/human-phenotype-ontology/blob/master/src/ontology/reports/hpodiff_hp_2021-06-13_to_hp_2021-08-02.xlsx) terms for cases with a Rady Children's Hospital Epic EHR were automatically extracted in four steps by natural language processing (NLP) of text fields:[16] (1) Clinical records were exported from the Epic EHR data warehouse, transformed into a compatible format (JSON), and loaded into CLiX ENRICH v.6.7 (CliniThink Ltd.). (2) A semi-automated query map was created, with HPO terms (and their synonyms) as the input and CLiX queries as the output. The HPO terms were passed through the CLiX encoding engine, resulting in creation of CLiX post-coordinated SNOMED CT (https://confluence.ihtsdotools.org/display/RMT/SNOMED + CT + January+2022+International+Edition + −+SNOMED + International+Release+notes) expressions for each recognized HPO term or synonym. Where matches were not exact, manual review was used to validate the generated CLiX queries. Where there was no match or incorrect matches, new content was added to the Clinithink SNOMED CT extension and terminology files to ensure appropriate matches between phenotypes in HPO and those in SNOMED CT. This was an iterative process that resulted in a CLiX query set that covered 60% (7706) of 12,786 HPO terms. (3) EHR documents containing unstructured data were passed through the NLP processing engine. The NLP processing engine read the unstructured text and encoded it in structured format as post-coordinated SNOMED CT expressions. These encoded data were then interrogated by the CLiX query technology (abstraction). To trigger an HPO query, the encoded data had to contain either an exact match or one of its logical descendants (exploiting the parent-child hierarchy of the SNOMED CT ontology), resulting in a list of HPO terms for each patient. EHR data for cases from partner hospitals was imported as machine-readable.pdf files to CLiX ENRICH v.6.7. In cases with more than one.pdf file, they were combined into a.zip file for upload to CLiX ENRICH. The NLP engine read the unstructured text and encoded it as HPO terms, resulting in a list of observed terms for each patient[49]. The analytic performance of NLP by CLiX ENRICH v.6.7 and v.6.5 was compared with manual chart review by two physician experts for ten test cases[16].

**Rapid diagnostic whole genome sequencing.** The standard clinical rWGS methods were DNA isolation from EDTA blood samples with the EZ1 DSP DNA Blood Kit (Qiagen, Cat. No. 62124), followed by library preparation with the polymerase chain reaction (PCR)-free KAPA HyperPrep kit (Roche, Cat. No. KK8505), and 2 × 101 nucleotide (nt) sequencing on NovaSeq 6000 instruments (Illumina, Cat. No. 20013850) with S1 flowcells, v.1 reagents, and standard recipe (Illumina, Cat. No. 20028319)[14]. The 19.5-h rWGS methods were library preparation from EDTA blood samples with Nextera DNA Flex Library Prep kits (Illumina, Cat. No. 20018705) and five cycles of PCR, 2 × 101 nt sequencing without indexing on NovaSeq 6000 instruments with S1 flowcells, v.1.0 reagents, and a custom recipe with accelerated cycle time (Illumina, Cat. No. 20012864), and

sequence alignment and nucleotide variant detection with the DRAGEN Platform (v.2.5.1, Illumina, Cat. No. 20060401)[16].

For 13.5-h rWGS, sequencing libraries were prepared directly from EDTA blood samples or five 3 mm² punches from a Nucleic Card Matrix dried blood spot (ThermoFisher, Cat. No. 4473977), without intermediate DNA purification, using magnetic bead-linked transposomes (DNA PCR-free Prep kit, Tagmentation, Illumina, Cat. No. 20041795)[23,24]. The length of each incubation step was maximally reduced from those in the manufacturer's protocol (Fig. 1). The shorter incubations normalized library output, which enabled simpler, faster measurement of library concentration with a KAPA Library Quantification Kit (Roche, Cat. No. 07960140001). 2 × 101 cycle sequencing-by-synthesis was performed on NovaSeq 6000 instruments (Illumina, Cat. No. 20013850) with a custom instrument run recipe with maximally reduced cycle time consistent with retention of sequence quality. Sequencing used SP flowcells and version 1.5 reagents (Illumina, Cat. No. 20040719), which were more cost effective and delivered better sequence quality than v.1.0 reagents. Sequences were aligned to human genome assembly GRCh37 (hg19), and variants identified and genotyped with the DRAGEN platform v.3.7.5 (Illumina). Automated variant interpretation was performed in parallel using MOON (InVitae), GEM (Fabric Genomics), and the Illumina TruSight Software Suite (TSS, Illumina)[16,33]. Inputs were the variant call file (vcf), list of observed HPO terms, and patient metadata (coded identifier, name, EHR number, ordering physician, date of birth, location, relationship to proband). All three software platforms (MOON, GEM, and TSS) generated a list of potential provisional diagnoses by sequentially filtering and ranking variants using decision trees, Bayesian models, neural networks, and natural language processing. The three software platforms ranked variants according to phenotypic match, pathogenicity, and rarity (Table S3). For generalizable, high throughput clinical use, each of these components was integrated with a custom laboratory information management system (LIMS, L7 Inc.) and custom analysis pipeline (Axolotl v.5.0, Rady Children's Institute for Genomic Medicine) that automated data transfers between steps.

**Measurement of analytic performance of rWGS.** The analytic performance of the new rWGS methods was compared with prior clinical rWGS methods in two reference DNA samples (NA12878, https://catalog.coriell.org/0/Sections/Search/Sample_Detail.aspx?Ref=NA12878, and NA24385, https://catalog.coriell.org/0/Sections/Search/Sample_Detail.aspx?Ref=NA24385&Product=DNA) using NIST gold standard variant sets for SNVs and indels (NISTv4.1, https://ftp-trace.ncbi.nlm.nih.gov/giab/ftp/release/), and SVs and CNVs (NISTv0.6, ftp://ftp-trace.ncbi.nlm.nih.gov/ReferenceSamples/giab/data/AshkenazimTrio/analysis/NIST_SVs_Integration_v0.6/) and Witty.er v0.3.4 (https://github.com/Illumina/witty.er/releases)[25,26,29].

**Gene and intervention curation.** 358 genes associated with 563 critical, childhood-onset illness with effective treatments were identified by literature review, subspecialist nomination and rapid precision medicine experience (Supplementary Data 1). Automated scripts were written to collect information about the gene, inheritance pattern, natural history and interventions from pubicly available information resources. Gene to disease mapping was done using OMIM (https://www.omim.org/) and Orphanet (https://www.orpha.net/consor/cgi-bin/Disease.php?lng=EN) mappings. Resources included OMIM, Orphanet, Clinical Trials (https://clinicaltrials.gov/ct2/home), ClinVar (https://www.ncbi.nlm.nih.gov/clinvar/), clinical trial registries including the Cochrane database (https://www.cochranelibrary.com/central/about-central), DrugBank v5.0 (https://go.drugbank.com/releases/latest), Gene (https://www.ncbi.nlm.nih.gov/gene), Genetic and Rare Disease Information Center (GARD) https://rarediseases.info.nih.gov/diseases, GeneReviews (https://www.ncbi.nlm.nih.gov/books/NBK1116/), Inxight:Drugs (https://drugs.ncats.io/substances), GHR (https://medlineplus.gov/genetics/gene/ghr/), MedGen (https://www.ncbi.nlm.nih.gov/medgen/), Medscape (https://reference.medscape.com/), NORD (https://rarediseases.org/for-patients-and-families/information-resources/rare-disease-information/), and PubMed (https://pubmed.ncbi.nlm.nih.gov/). Scripts were also written to identify published literature relating to each condition and identify pertinent treatments (Genomenon Inc. Rancho Biosciences, Epam). Publications were included if they mentioned the condition, the specific variant identified, and a clinical intervention used to treat the condition. Intervention lists for each gene-condition association were curated manually for relevance and specificity to the intensive care setting.

**Expert review panel.** The list of interventions for each gene-condition association was adjudicated by a group of expert reviewers. Reviewers were experts in the fields of clinical and biochemical genetics. Five reviewers in total were recruited for the first stage of interface development. Software for intervention review was developed using the RedCap interface (RedCap, https://redcap.radygenomiclab.com/redcap_v10.6.3/DataEntry/record_status_dashboard.php?pid=62, Figs. S4–S7), and reviewers were able to login via a web portal in order to review genes that had been curated by a combination of AI and manual curation. Expert consensus on curated interventions was required for the inclusion on the final user interface, as illustrated in Fig. 2. In Phase 1, reviewers were provided with a prototype set of 10 genes in order to test the reviewer interface, after which a concordance analysis was performed and the RedCap interface was extensively revised in response to reviewer feedback. The reviewers then reviewed

the same 10 gene set again, with an additional 5 genes associated with pre-selected retrospective cases. Reviewers chose whether to retain or delete previously curated interventions, and indicated in what age group the intervention may be initiated, in what time frame after diagnosis the intervention would optimally be initiated, contra-indications, efficacy, and level of evidence available in support of the intervention (Box 1). A set of core inclusion and exclusion criteria for interventions was drafted and revised by the group, as detailed in the Supplementary Materials. After initial review of the 15 gene pilot set, the interventions on which consensus was not reached were discussed in roundtable discussion. In Phase 2, reviewers were split into pairs, and each gene had one reviewer perform a primary review, and a second reviewer perform a secondary review (Fig. 2). Any disagreements between the primary and secondary expert review were again discussed in the roundtable meeting with all reviewers, and only interventions that reached full consensus were included. The final list of inter-ventions was collated after full consensus had been reached between all five reviewers. As a final quality control and assurance step, an independent expert performed a final quality check for each gene before moving it to the user interface pipeline.

**User interface development and integration into automated pipeline**. A web resource integrated the GTRx information resources and the adjudicated inter-ventions (http://gtrx.rbsapp.net/). The user interface for GTRx was developed in partnership with Rancho Biosciences. Automated scripts integrated the electronic acute disease management support system into MOON (Diploid), GEM (Fabric Genomics), and the Illumina TruSight Software Suite (Illumina). This provided an automated link to treatment guidance once a provisional genetic diagnosis was reached by the variant curation tool. The provisional management plan auto-matically generated by GTRx for each of the four retrospective cases were checked by a lab director and a clinician for accuracy.

**Reporting summary**. Further information on research design is available in the Nature Research Reporting Summary linked to this article.

## Data availability

Source data are provided with this paper. The processed patient data generated in this study have been deposited in the Longitudinal Pediatric Data Resource (LPDR) under accession code nbs000003.v1.p at https://nbstrn.org/. LPDR data are available to eligible investigators within 5 business days following registration at https://nbstrn.org/registration subject to the terms and conditions listed on the registration page. The raw patient data are protected and not available due to data privacy and confidentiality laws. Anonymized and pseudonymized patient data generated in this study, subject to the terms of informed written consent documents, and state and federal laws, are provided in the Supplementary Information/Source Data file. Non-human subjects data generated in this study are provided in the Supplementary Information/Source Data file. NIST data used in this study are available at https://ftp-trace.ncbi.nlm.nih.gov/giab/ftp/release/, and ftp://ftp-trace.ncbi.nlm.nih.gov/ReferenceSamples/giab/data/AshkenazimTrio/analysis/NIST_SVs_Integration_v0.6/. Raw non-human subjects data generated in this study data are available from Dr. Yan Ding (YDing@rchsd.org) or Dr. Kevin Hall (KHall@illumina.com). Source data are provided with this paper.

## Code availability

Witty.er is available at https://github.com/Illumina/witty.er. InterVar is available at https://github.com/WGLab/InterVar. GTRx is available at https://gtrx.radygenomiclab.com/ and code is available from Christian Hansen (chansen@rchsd.org) and at https://github.com/rao-madhavrao-rcigm/gtrx. CLIXEnrich is available from CliniThink (info@clinithink.com). Moon is available from Invitae or Diploid (info@diploid.com). The DRAGEN Platform and the Illumina TruSight Software Suite are available from Illumina (Shyamal Mehtalia, smehtalia@illumina.com, www.illumina.com). OPAL and GEMS are available from Fabric Genomics (info@fabricgenomics.com). The RCIGM portal, Axolotl pipeline, and L7 LIMS are available from Danny Oh (doh@rchsd.org) and at https://github.com/rao-madhavrao-rcigm/gtrx. The GTRx REDCap instance are available from Christian Hansen (chansen@rchsd.org) and at https://github.com/rao-madhavrao-rcigm/gtrx. The KNIME pipeline is available from Sebastien Lefebvre (sebastien.lefebvre@alexion.com).

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

## Acknowledgements

We thank Jared Vogt and Matt DeFinis for help in data export from the Rady EPIC EHR, Paula Gray, and Scott Voigts, Santosh Arunuri and Brad Soileau for computing infrastructure support. This work was supported by gifts from Ernest and Evelyn Rady, grant U19HD077693 from NICHD and NHGRI to S.F.K., grant U01TR002271 from NCATS to J.M. Davis and J.L. Maron (with sub-award to S.F.K.), and grant UL1TR002550 from NCATS to E.J. Topol (with sub-award to S.F.K.), and funds from Rady Children's Hospital.

## Author contributions

S.F.K., J.R., D.D., S.L. conceived and designed the study. S.L. developed the KNIME pipelines. C.H. developed the GTRx REDCap instance. C.K. provided project management for the study. M.O., D.D., M.J.W., L.D.S., G.S., R.M., A.F., and J.P. reviewed the disorders and treatments included in GTRx. M.O., S.R., K.C., N.V., LVDK, Y.D., Y.K., Z.Z., S.B., S.C., M.W., M.R., F.V., J.A., E.F., C.S.R., S.W., CH S.G., G.K., A.O., J.L., S.N., S.L., J.W., V.T., L.F., and K.H. generated, analyzed, and interpreted the data. AK.N., M.S., J.C., G.K., N.G.C., K.P., K.K., E.S., and M.N. provided clinical care for the patients presented herein. M.S.W., S.C., K.E., L.G., and S.A.N. performed clinical analysis and interpretation of genome sequences. Y.D. and L.V.D.K. performed clinical genome sequencing. J.L. provided genetic counseling. K.W. provided clinical genetics consultation. M.O. and S.F.K. wrote the manuscript. All of the authors critically revised the manuscript. S.S.M., C.H., S.L., K.K.C, Y.H.K., Z.Z., S.B., D.T., S.G., N.V., and A.O. provided bioinformatic support. M.G.R., F.M.D.L.V., J.A., E.F., C.S.R, and S.W. developed Fabric Enterprise and Fabric GEM and assisted with the Fabric GEM

analysis. C.M.K, B.R.L., T.K.M., J.W., V.S.T., L.F., S.S.L., Y.H.S., H.G., R.L., D.J., A.G., P.D., A.C., M.L., J.A.R., and K.P.H. developed the modified Illumina sequencing-by-synthesis chemistry, modified tagmentation library preparation methods, faster SP flowcell recipe, and Illumina automated interpretation software. C.A.H., M.J.O., S.F.K., K.P., K.K., N.G.C., and E.S. performed research analyses. J.S.R., S.F.K, S.L., N.V., K.P.H., and T.D. supervised the study.

## Competing interests

M.G.R., F.D.L.V., C.S.R., J.A., S.W., and E.F. are employees and stockholders of Fabric Genomics, Inc. and have equity holdings and stock options. T.D., J.R., and S.L. are employees of Alexion Pharmaceuticals, Inc. and have equity holdings and stock options. K.P.H., J.A.R., M.L., A.C., P.D., A.G., D.T., D.J., R.L., H.G., S.S.L., Y.H.S., L.F., C.M.K., B.R.L., T.K.M., S.S.M., J.W., and V.S.T. are employees of Illumina, Inc. and have equity holdings and stock options. D.D. received funding from Biomarin (consultant for Peg-valiase trials), Audentes Therapeutics (Scientific Advisory Board), and Ichorion Therapeutics (consultant for mitochondrial disease drugs). M.B. owns stock in Codified Genomics and is a consultant for Baebies Inc. Related to the current work, S.F.K. and S.L. filed provisional patent application 63/209,797, entitled "Method and system for improved management of genetic diseases" with the U.S. Patent and Trademark Office. The remaining authors have no competing interests.

## Additional information

[1]Rady Children's Institute for Genomic Medicine, San Diego, CA 92123, USA. [2]Rady Children's Hospital, San Diego, CA 92123, USA. [3]Alexion Pharmaceuticals, Inc., Boston, MA 02210, USA. [4]Illumina, Inc., San Diego, CA 92122, USA. [5]Keck Graduate Institute, Claremont, CA 91711, USA. [6]Department of Pediatrics, University of California San Diego, San Diego, CA 92093, USA. [7]Department of Neuroscience, University of California San Diego, San Diego, CA 92093, USA. [8]Fabric Genomics, Inc., Oakland, CA 94612, USA. ✉email: skingsmore@rchsd.org

