## [Peer Review File · Nature Communications]

REVIEWER COMMENTS

Reviewer #1 (Remarks to the Author):

The manuscript reports GTRx, an automated virtual system for genetic disease diagnosis and acute management guidance for ill children in intensive care units (ICU). The system includes 563 severe genetic diseases with effective treatments identified through an adjudication process. It includes a genetic disease diagnosis component and an acute management guidance component to support the use of rapid whole genome sequencing (rWGS) technologies in clinic. The system has a potential for optimal acute treatment for children with rapidly progressive genetic diseases.

The system was demonstrated with a great potential to help the acceleration of genetic disease diagnosis and management. However, this reviewer has some major concerns.

1. The knowledge resource utilized by the system are currently curated from various knowledge resources. It is not clear if the system will automatically keep up to date. If not, the scalability may be limited.

2. The use of Natural Language Processing techniques seems to be the key in helping with disease diagnosis. In general, when genetic diseases are suspected, the HPO terms will start to appear but not before the ordering of rWGS. It is not clear how exactly the temporal information associated with clinical documentation is considered in the diagnosis part.

3. It is not clear how exactly the system interfaces with the EHR and the sequencers. Are standards adopted?

4. Does the research itself reproducible and FAIR?

Reviewer #2 (Remarks to the Author):

Owen et al. summarize a comprehensive program to perform rapid genome sequencing, interpret results, and provide decision support information to clinicians. This is a tremendous amount of work, involving a very large team effort for a very complicated set of processes, so the authors are to be commended for their efforts.

There are several noteworthy aspects of this work.

1. The authors have modified molecular protocols to speed up the process of data generation as much as possible, and the results are impressive in terms of the time elapsed from sample collection to diagnosis, albeit based on small numbers.

2. They not only focused on generating rapid sequence data, but developed (or adapted) a comprehensive set of tools designed to facilitate rapid interpretation and reporting. The reported improvement in rapid calling of CNVs represents a significant step forward (although the details/validity were not evaluated by this reviewer).

3. This reviewer agrees with the authors that the interpretation and reporting process is the biggest bottleneck to overcome, and their approach attempts to address this in a comprehensive way. (e.g., L1ne 376: “manual interpretation and reporting are becoming the largest component of the expense of diagnostic rWGS.”)

5. Although the authors did not specifically address cost-effectiveness, there is the potential that the automation developed for this process could make a significant impact on cost-effectiveness of rapid genome sequencing since a large portion of the cost is related to personnel for interpretation and reporting.

5. The total time is particularly noteworthy. This author group has already set impressive records for turnaround time, and the present approach, if it is reliable and valid in the same proportion of cases when assessed at production scale, represents a significant advance even compared to those already impressive accomplishments.

This work is likely to be of significant interest in the medical genetics, neonatology, and pediatrics communities in general. The impact could potentially be great for patients with hundreds/thousands of rare genetic disorders. As mentioned, although these authors did not specifically address cost-effectiveness, the potential for cost-effectiveness is readily apparent and therefore this

approach is likely to be of high interest to healthcare systems/payers in addition to the medical community.

There are certain aspects of the presentation that should be described in more detail. This reviewer cannot comment specifically about the natural language processing and other AI methods. However, this reviewer has extensive experience as a clinical geneticist and molecular geneticist working in clinical laboratories, and suspects that the general reader will not be able to identify/evaluate some of the nuances of this presentation.

1. The validation was performed on four retrospectively selected, and one prospectively selected, cases. These cases were not randomly selected. This could be a potential limitation/source of bias, and the Discussion currently does not discuss any potential limitations of the study, but should. These methods would be expected to work best (and most quickly) with certain types of inheritance patterns (i.e., recessive disorders where there are two variants in one gene, or dominant/X-linked disorders that are de novo). For example, choosing cases with a severe neonatal phenotype and apparent recessive inheritance (older sibling affected and parents asymptomatic) is already a well documented way to have a very high chance to identify the causative gene/variant by exome/genome sequencing, but would only work if the causative gene is part of the pre-curated set of genes. Please discuss.

2. Likewise, the authors chose gene-disease pairs where there is not locus heterogeneity, and there is a clear genotype-phenotype correlation (metabolic disorders caused by one enzyme in 3 of 4 cases). This type of reasoning for selection of these specific cases makes sense for a proof of concept, but may result in lower rates of success in real world application. Please discuss that, and also the reason for selecting only 4 cases (and how other types of cases may not be as straightforward), in the Discussion.

3. Also, the speed of analysis is aided by trio sequencing, but nowhere in the manuscript is it described specifically that samples were processed as a trio (at least by searching for “trio” and “parent,” and it’s also not part of the flow diagram). They must have been trios in order to have confirmation of de novo status (for two of the cases) within the ~13 hours. For example, de novo status for the missense in SCN2A greatly helps the quick identification of the purported causative variant. This variant is not in ClinVar, and - as a novel missense variant - would have been classified uncertain otherwise. This should be described more explicitly. It is important that the reader knows that this requires trio sequencing in some/many cases.

4. It would be difficult for a typical clinical laboratory to implement these methods without much more explanation of how the software works, and also more explanation about how the group review process of the biochemical geneticists was conducted. For example, that is a time-consuming effort that took place prior to processing the samples. It would help the reader to have some idea how much time the group needed to spend curating each condition, as that affects scalability of the process.

5. It will be difficult for the typical reader to have a clear picture of the precision and recall metrics related to phenotypic features presented in table S1. The formulas for determining these are clear,

but it is not intuitive how the NLP method would result in false positives. Does FP here mean attributions of phenotypic features that the algorithm assumes are attributable to the condition, and yet they are not? If so, why does the algorithm attribute phenotypic features incorrectly to a condition if they are not listed in association with that condition in some type of database (e.g., OMIM) that was used to train the algorithm?

6. For readers without the expertise in NLP methods, it would help the general reader for the authors to provide a benchmark of what is considered “good performance” in terms of precision and recall. Apologies if this was described somewhere.

7. It is very interesting that five clinical geneticists agreed upon 189 of the first 190 treatments. However, it is quite hard for the reader to get a sense of how burdensome that process was. Could the authors describe more about average time per curation?

8. (Line 359) “unstable infants” is subjective. Could you provide more clarification? Is this related to suspected metabolic disorders, or would it include patients with pulmonary or circulatory compromise as an isolated finding. For example, extremely low birthweight infants are unstable, but would not be suspected of having a genetic syndrome. I am thinking more of the general audience here; it would help them to know “who looks like a good candidate for this testing.”

9. For the thiamine-responsive seizures case, there are very few details about the case (“At six months of age, he was thriving”). I suggest saying something more objective about the outcome such as something regarding developmental milestones being met, etc.

Minor specific edits.

(Table S1 and S2) Since this article is directed at more of a general audience, I suggest putting “n/a” in the column for variant 2 when the condition is autosomal dominant. This is obvious to a geneticist, but may not be to a general audience. Alternatively, just explain this in the footnotes of the table.

(Table S6) is labelled as “Table 6”

Reviewer #3 (Remarks to the Author):

Reviewer Critique

In a study, entitled " Genome-to-Treatment: A virtual, automated system for population-scale diagnosis and acute management guidance for genetic diseases in 13.5 hours ", Dr. Owen and colleagues outline the important role of rapid whole genome sequencing with an automated analysis pipeline to establish rapid diagnosis and treatment of rare genetic diseases that have effective treatments as many of those progress rapidly to severe morbidity or mortality if not addressed immediately. They emphasize that front-line physicians are often unfamiliar with these diseases or treatments, hence the need to establish a workflow to follow. The authors describe Genome-to-Treatment (GTRx), an automated, virtual system for genetic disease diagnosis and acute management guidance for ill children in intensive care units. They present examples where diagnosis was achieved in 13.5 hours by sequencing library preparation directly from blood, faster whole genome sequencing (WGS) and informatic analysis, natural language processing of electronic health records and automated interpretation. Upon literature review, they identified 563 severe genetic diseases with effective treatments (drugs, devices, diets, and surgeries) based on clinician nomination by 5 experts in the context of their WGS experience.

The team agreed upon 189 of the first 190 treatments proposed. The authors integrated 10 genetic disease information resources, and electronically linked them and the adjudicated treatments to each automated diagnostic result (<http://gtrx.rbsapp.net/>). This system had superior analytic performance for single nucleotide, insertion-deletion, structural and copy number variants and the author present correct diagnoses and acute management guidance in four retrospective patients. Prospectively, an infant with encephalopathy was diagnosed in 13.5 hours, received effective treatment immediately, and had a good outcome. The authors conclude that GTRx will facilitate broad implementation of optimal acute treatment for children with rapidly progressive genetic diseases by front-line intensive care unit physicians. While this study present an impressive and effective accomplishments by the team in rapidly uncovering genetic variants and linking them to existing therapies and as such of potential interest to the readership of Nature Communications, a significant component of the manuscript involves mining of existing data with recommendations and treatment guidelines that are more in keeping with a review process and this reviewer wonders if the work would not be more appropriate with some modification for a high impact review paper (such as Nat Rev Genet). I have outlined my concerns in the comments below:

Major Comments:

1. The authors generated sequence data in an efficient way and use commercial data mining tools and algorithms (InVitae MOON, Fabric GEMS and Illumina TruSight Software Suite, (TSS)), to annotate the resulting variants in search for a causative variant that confirms diagnosis and would dictate potential therapy. This is of course what numerous genome centers and institutes do on a regular basis and would be helpful if the authors would delineate more specifically what the innovative component is that is unique to their GTRx.

2. Similarly, all the data mining work for the known 563 severe genetic diseases with effective treatments is independent of any new data being generated and the innovative contribution here should be brought better to light and avoid commercial influence.
3. It would seem important to address the gap between almost 6000 known diseases and the roughly 500 that have potential therapies – what are the authors doing about the 5500 diseases that have no therapies and what are they recommending for this unmet need.
4. The authors emphasize the strong evidence that exist in support of the notion that diagnosis of genetic diseases by rWGS improves outcomes of infants and children in intensive care units and the approach has been implemented in several countries and multiple states in the US, which this reviewer endorses. The key issue is how best to convey this message to medical practices that are not currently implementing rWGS for their neonatal care, hence the suggestion of a white paper or high impact review manuscript in journals such as Nat Rev Genet.
5. The authors state that another innovation of the automated system they describe is the ability to diagnose genetic diseases associated with all major classes of genomic variants. This is perhaps a bit of an overstatement as is common practice so not terribly innovative and perhaps a different wording should be used to describe this function.

Minor comments:

1. Important to attenuate any tone driving commercialization of this approach and focus on the science.

Reviewer #1:

1. The knowledge resource utilized by the system are currently curated from various knowledge resources. It is not clear if the system will automatically keep up to date. If not, the scalability may be limited.

Response: Since submission we increased the number of conditions reviewed from 100 to 449. We will continue until all 563 diseases are completed (Figure 3). We addressed the limitations of GTRx and plans for scalability and sustainability by the addition of the following paragraph to the discussion:

“Version 1 of GTRx, described herein, was limited to genetic diseases of known molecular cause, that can be diagnosed by rWGS, can lead to ICU admission in infancy, and have effective treatments. During development, we realized that not all genetic diseases that meet these criteria were represented in the set of 563. Furthermore, the literature related to known genetic diseases and treatments is continually being augmented. While pediatric geneticists were optimal subspecialists for initial review of disorders and interventions, there are many that would benefit from additional sub- and super-specialist review. We plan to address these limitations in future versions of GTRx, with expert, open, community-based, ongoing review. In addition, recent evidence supports the use of rWGS for genetic disease diagnosis and management guidance in older children in PICUs. It is desirable to include these conditions in future versions. There are several, additional, complementary information resources that would enrich GTRx, such as ClinGen, the Genetic Test Registry, and Rx-Genes. Finally, there are a large number of clinical trials of new interventions for infant-onset, severe genetic disorders, particularly genetic therapies. For disorders without current effective treatments it is desirable to include links to enrollment contacts for those clinical trials.”

2. The use of Natural Language Processing techniques seems to be the key in helping with disease diagnosis. In general, when genetic diseases are suspected, the HPO terms will start to appear but not before the ordering of rWGS. It is not clear how exactly the temporal information associated with clinical documentation is considered in the diagnosis part.

Response: Yes, diagnostic variant interpretation is guided by the observed phenotypes in the patient. We have clarified the temporal association of HPO term extraction during diagnostic interpretation in the Results as follows:

“Firstly, the patients’ phenotypic features were extracted from non-structured text fields in the electronic health record (EHR) using natural language processing (NLP, Clinithink Ltd.) through the date of enrollment for WGS.¹⁶.... Secondly, for each patient, the extracted HPO terms observed in the patient at time of enrollment were compared with the known HPO terms for all ~7,000 genetic diseases with known causative loci.¹”

3. It is not clear how exactly the system interfaces with the EHR and the sequencers. Are standards adopted?

Response: We have revised Figure 1B to show more clearly how the system interfaces with the EHR and sequencers and the standards that have been adopted (such as HL7/FHIR and HPO terminology). We have clarified this in the text of the results as follows:

“First, we simplified ordering of rWGS. Orders are placed directly through the Epic EHR (Figure 1). The test order and patient metadata is transferred from the EHR to a custom ordering portal.”

“For generalizable, scalable clinical use, each of these components (sample accessioning, library preparation, library quality assessment, sequencing and variant calling) was integrated with a custom laboratory information management system and custom analysis pipeline (Enterprise Science Platform, L7 Informatics) that automated data transfers between steps.”

Yes, the system adheres to the technical standards for clinical genomic sequencing. We have clarified this in the discussion:

“The system adheres to the technical standards developed by the ACMG for diagnostic genomic sequencing.”

Rehder C, Bean LJH, Bick D, Chao E, Chung W, Das S, O'Daniel J, Rehm H, Shashi V, Vincent LM; ACMG Laboratory Quality Assurance Committee. Next-generation sequencing for constitutional variants in the clinical laboratory, 2021 revision: a technical standard of the American College of Medical Genetics and Genomics (ACMG). *Genet Med.* 2021 Aug;23(8):1399-1415.

4. Does the research itself reproducible and FAIR?

Response: Yes, the system generates results reproducibly. This is shown in Table 1. We have clarified this in the Table legend as follows:

“Table 1. Analytic performance, reproducibility, and duration of the major steps in automated diagnosis of genetic diseases by accelerated rWGS. Analytic and diagnostic reproducibility were examined for sample 362 from 19.5-hour rWGS (16), reference samples NA12878 and NA24385, and four retrospective samples/diagnoses (AG928/Hereditary fructose intolerance (compound heterozygous, pathogenic (P) SNVs in aldolase B [ALDOB c.448G>C, c.524C>A]); AG366/Ornithine transcarbamylase deficiency (hemizygous, de novo, P, SNV in ornithine transcarbamylase [OTC c.275G>A]); AF414/Propionic acidemia (homozygous, likely pathogenic (LP) indel in α -subunit of propionyl-CoA carboxylase [PCCA c.1899+4_1899+7del]); AI003 /Developmental and epileptic encephalopathy 11 (heterozygous, de novo, LP SNV in the $\alpha 2$ -subunit of the voltage-gated sodium channel [SCN2A c.4437G>C]). Reproducibility was also shown in three prospective samples AH638, CSD59F and CSD709, which received rWGS both with the novel 13.5-hour method (Herein) and standard, clinical rWGS (Std). 1⁰/2⁰ analysis time: Conversion of raw data from base call to FASTQ format, read alignment to the reference genomes and variant calling. Tertiary analysis: Time of automated interpretation to provisional diagnosis (most rapid of three systems run in parallel (MOON, Illumina TruSight Software Suite and GEM). SV and CNV detection methods: MC: Manta and CNVnator; : DRAGEN v.3.7. D3.5.3: DRAGEN v.3.5.3. MIM: Mendelian inheritance in man. Nt: Nucleotides.”

GTRx adheres to the FAIR principles elaborated in Reference 56 (Atalaia, A. et al. A guide to writing systematic reviews of rare disease treatments to generate FAIR-compliant datasets: building a Treatabolume. *Orphanet J. Rare Dis.* 15, 206, 2020) and 83 (Wilkinson MD, Dumontier M, Aalbersberg IJ, Appleton G, Axton M, Baak A, Blomberg N, Boiten JW, da Silva Santos LB, Bourne PE, Bouwman J, Brookes AJ, Clark T, Crosas M, Dillo I, Dumon O, Edmunds S, Evelo CT, Finkers R, Gonzalez-Beltran A, Gray AJ, Groth P, Goble C, Grethe JS, Heringa J, 't Hoen PA, Hooft R, Kuhn T, Kok R, Kok J, Lusher SJ, Martone ME, Mons A, Packer AL, Persson B, Rocca-Serra P, Roos M, van Schaik R, Sansone SA, Schultes E, Sengstag T, Slater T, Strawn G, Swertz MA, Thompson M, van der Lei J, van Mulligen E, Velterop J, Waagmeester A, Wittenburg P, Wolstencroft K, Zhao J, Mons B. The FAIR Guiding Principles for scientific data management and stewardship. *Sci Data.* 2016 Mar 15;3:160018).

We have clarified this in the Results as follows:

“The retained interventions and qualifying statements were incorporated into the GTRx information resource as a prototypic acute management guidance system for genetic diseases that meets FAIR principles^{56,83}”

Reviewer #2 (Remarks to the Author):

Owen et al. summarize a comprehensive program to perform rapid genome sequencing, interpret results, and provide decision support information to clinicians. This is a tremendous amount of work, involving a very large team effort for a very complicated set of processes, so the authors are to be commended for their efforts.

There are several noteworthy aspects of this work.

1. The authors have modified molecular protocols to speed up the process of data generation as much as possible, and the results are impressive in terms of the time elapsed from sample collection to diagnosis, albeit based on small numbers.
2. They not only focused on generating rapid sequence data but developed (or adapted) a comprehensive set of tools designed to facilitate rapid interpretation and reporting. The reported improvement in rapid calling of CNVs represents a significant step forward (although the details/validity were not evaluated by this reviewer).
3. This reviewer agrees with the authors that the interpretation and reporting process is the biggest bottleneck to overcome, and their approach attempts to address this in a comprehensive way. (e.g., Line 376: “manual interpretation and reporting are becoming the largest component of the expense of diagnostic rWGS.”)
4. Although the authors did not specifically address cost-effectiveness, there is the potential that the automation developed for this process could make a significant impact on cost-effectiveness of rapid genome sequencing since a large portion of the cost is related to personnel for interpretation and reporting.
5. The total time is particularly noteworthy. This author group has already set impressive records for turnaround time, and the present approach, if it is reliable and valid in the same proportion of cases when assessed at production scale, represents a significant advance even compared to those already impressive accomplishments.

This work is likely to be of significant interest in the medical genetics, neonatology, and pediatrics communities in general. The impact could potentially be great for patients with hundreds/ thousands of rare genetic disorders. As mentioned, although these authors did not specifically address cost-effectiveness, the potential for cost-effectiveness is readily apparent and therefore this approach is likely to be of high interest to healthcare systems/payers in addition to the medical community.

There are certain aspects of the presentation that should be described in more detail. This reviewer cannot comment specifically about the natural language processing and other AI methods. However, this reviewer has extensive experience as a clinical geneticist and molecular geneticist working in clinical laboratories, and suspects that the general reader will not be able to identify/evaluate some of the nuances of this presentation.

1. The validation was performed on four retrospectively selected, and one prospectively selected, cases. These cases were not randomly selected. This could be a potential limitation/source of bias, and the Discussion currently does not discuss any potential limitations of the study but should.

Response: We have expanded the reported prospective cases from one to three. We revised the Results to include details of the two new prospective cases and added the second case to Figure 4. We have started to use these methods for selected cases, and added the corresponding following sentences to the discussion:

“In clinical production in three cases, we have found that these new methods have reduced this by a factor of two.”

We also published a manuscript on the diagnostic performance of the GEM AI method for variant interpretation and have referenced it in the discussion as follows:

“We recently evaluated the diagnostic performance of GEM, the automated interpretation system, in 193 children with suspected genetic diseases³⁹. In 92% of cases, GEM ranked the correct gene and variant in the top two calls, including structural variant diagnoses.”

We have explicitly addressed the reviewer’s concern by modifying the following sentences in the Discussion:

“However, to date the system has been evaluated only in four retrospective and six prospective cases. Further studies are needed for clinical validation, such as reproducibility, performance with all patterns of inheritance and all pathogenic variant types, examination of the relative diagnostic performance of automated methods compared with traditional manual interpretation, and to understand the proportion of edge cases.”

The Discussion does discuss other potential limitations as follows:

"It should be noted, however, that recall (sensitivity) for SVs and CNVs remain a weakness of short read sequencing (range 49.3% - 87.9%). The consequences of this for genetic disease diagnostic sensitivity is unknown. Studies are needed to compare the diagnostic performance of these methods versus hybrid methods with short read sequencing and complementary technologies, such as long-read sequencing and optical mapping."^{73,74}

We have also added the following paragraph regarding the limitations of GTRx to the Discussion:

"Version 1 of GTRx, described herein, was limited to genetic diseases of known molecular cause, that can be diagnosed by rWGS, can lead to ICU admission in infancy, and have effective treatments. During development, we realized that not all genetic diseases that meet these criteria were represented in the set of 563. Furthermore, the literature related to known genetic diseases and treatments is continually being augmented. While pediatric geneticists were optimal subspecialists for initial review of disorders and interventions, there are many that would benefit from additional sub- and super-specialist review. We plan to address these limitations in future versions of GTRx, with ongoing, expert, open, community-based review. In addition, recent evidence supports the use of rWGS for genetic disease diagnosis and management guidance in older children in PICUs. It is desirable to include these conditions in future versions. There are several, additional, complementary information resources that would enrich GTRx, such as ClinGen, the Genetic Test Registry, and Rx-Genes⁸⁵⁻⁸⁷. Finally, there are many clinical trials of new interventions for infant-onset, severe genetic disorders, particularly genetic therapies. For disorders without current effective treatments it is desirable to include links to enrollment contacts for those clinical trials."

2. These methods would be expected to work best (and most quickly) with certain types of inheritance patterns (i.e., recessive disorders where there are two variants in one gene, or dominant/X-linked disorders that are de novo). For example, choosing cases with a severe neonatal phenotype and apparent recessive inheritance (older sibling affected and parents asymptomatic) is already a well documented way to have a very high chance to identify the causative gene/variant by exome/genome sequencing, but would only work if the causative gene is part of the pre-curated set of genes. Please discuss.

Response: As noted above, we have added two new prospective cases, one of which was a heteroplasmic mitochondrial variant, and cited a recently published manuscript that evaluated the diagnostic performance of the GEM AI method for variant interpretation in a broad set of presentations and causative variant types.

I believe that we have explicitly addressed the reviewer's concern by modifying the following sentences in the Discussion:

"However, to date the system has been evaluated only in four retrospective and six prospective cases. Further studies are needed for clinical validation, such as reproducibility, performance with all patterns of inheritance and all pathogenic variant types, examination of the relative diagnostic performance of automated methods compared with traditional manual interpretation, and to understand the proportion of edge cases."

2. Likewise, the authors chose gene-disease pairs where there is not locus heterogeneity, and there is a clear genotype-phenotype correlation (metabolic disorders caused by one enzyme in 3 of 4 cases). This type of reasoning for selection of these specific cases makes sense for a proof of concept but may result in lower rates of success in real world application. Please discuss that, and the reason for selecting only 4 cases (and how other types of cases may not be as straightforward), in the Discussion.

Response: I believe that our responses above, have addressed this issue and have clearly made the point that further work must be undertaken before AI-alone methods can be used in routine clinical diagnostics.

3. Also, the speed of analysis is aided by trio sequencing, but nowhere in the manuscript is it described specifically that samples were processed as a trio (at least by searching for “trio” and “parent,” and it’s also not part of the flow diagram). They must have been trios in order to have confirmation of de novo status (for two of the cases) within the ~13 hours. For example, de novo status for the missense in SCN2A greatly helps the quick identification of the purported causative variant. This variant is not in ClinVar, and - as a novel missense variant - would have been classified uncertain otherwise. This should be described more explicitly. It is important that the reader knows that this requires trio sequencing in some/many cases.

Response: Analysis speed is actually slowed by trio sequencing. The GEM artificial intelligence tool performs as well with singleton and trio samples (reference 39). The 13.5-hour method requires SP flowcells on the Illumina NovaSeq instrument. This flowcell generates ~150 GB of DNA sequence (~300 GB per 2-flowcell run). To achieve 13.6 hours, we run a ~50-fold singleton genome. Bioinformatic time increases linearly with the number of genomes processed, which would delay results. We have clarified this in the Table 1 legend as follows “.....which received rWGS both with the novel, singleton 13.5-hour method (Herein) and standard, singleton or trio, clinical rWGS (Std).” We have also inserted the word “singleton” in Figure 1A.

The reviewer is correct that trio testing is needed to confirm the de novo occurrence of variants in dominant disorders, which is sufficient evidence to promote pathogenicity classification from VUS to LP. In practice, however, for a variety of reasons, trio samples are often unavailable in time for GEM interpretation. In cases where an infant is critically ill and the provisional diagnosis is a disorder for which effective treatment is available, and where the delay in time to confirm de novo occurrence may lead to a poor outcome, we provisionally report suspicious VUS. We have a pre-investigational device exemption opinion from the FDA that this constitutes non-significant risk in such cases. In practice, this is limited to cases where there is very good correspondence of the clinical phenotype and that of the provisional diagnosis and the variant must either be novel or extremely rare in gnomAD.

4. It would be difficult for a typical clinical laboratory to implement these methods without much more explanation of how the software works, and also more explanation about how the group review process of the biochemical geneticists was conducted. For example, that is a time-consuming effort that took place prior to processing the samples. It would help the reader to have some idea how much time the group needed to spend curating each condition, as that affects scalability of the process.

Response: For the Genome-to-Treatment management guidance system, primary review of interventions for a disorder takes 1-5 hours, and secondary review about one hour. We have added a sentence to this effect to the Results (line 378-380). Upon publication, GTRx will be made freely available. Development of GTRx is “offline” with regard to processing of patient samples for diagnosis.

5. It will be difficult for the typical reader to have a clear picture of the precision and recall metrics related to phenotypic features presented in table S1. The formulas for determining these are clear, but it is not intuitive how the NLP method would result in false positives. Does FP here mean attributions of phenotypic features that the algorithm assumes are attributable to the condition, and yet they are not?

Response: We described the details of the false positive clinical features identified by NLP in a previous manuscript that is cited in the text [16]. From that reference: “The principal reasons for false positives were as follows: (i) incorrect CLiX encoding (n = 89, 38% of 237 phenotypic features) due to misinterpreted context (n = 31), unrecognized headings (n = 23), incorrect acronym expansion (n = 21), incorrect interpretation of a clinical word (n = 8), or incorrectly attributed finding site for disease (n = 6); (ii) ambiguity of source text (unrecognized or incorrect syntax, abbreviations, acronyms, or terminology; n = 46, 19% of 237); (iii) incongruity among SNOMED CT, HPO, and clinical acumen (n = 20, 8%); (iv) failure to recognize a pasted citation as nonclinical text (n = 68, 29%); and (v) incorrect query logic (n = 14, 6%).” We have added a sentence to this effect to the results (lines 190 - 191) as follows:

“The performance of NLP in extraction of clinical features from EHRs and reasons for identification of false positive clinical features have been previously described¹⁶.”

6. If so, why does the algorithm attribute phenotypic features incorrectly to a condition if they are not listed in association with that condition in some type of database (e.g., OMIM) that was used to train the algorithm?

Response: The algorithm does only attribute phenotypic features that have been associated with that condition in an extant reference database, such as OMIM, Orphanet, or GARD.

6. For readers without the expertise in NLP methods, it would help the general reader for the authors to provide a benchmark of what is considered “good performance” in terms of precision and recall. Apologies if this was described somewhere.

Response: We described the performance of NLP in clinical feature detection in EHRs in detail in a previous manuscript that is cited in the text [16]. We have added a sentence to this effect to the results (lines 190 - 191) as follows:

“The performance of NLP in extraction of clinical features from EHRs and reasons for identification of false positive clinical features have been previously described¹⁶.”

7. It is very interesting that five clinical geneticists agreed upon 189 of the first 190 treatments. However, it is quite hard for the reader to get a sense of how burdensome that process was. Could the authors describe more about average time per curation?

Response: For the Genome-to-Treatment management guidance system, primary review of interventions for a disorder takes 1-5 hours, and secondary review about one hour. We have added a sentence to this effect to the Results (line 378-380).

8. (Line 359) “unstable infants” is subjective. Could you provide more clarification? Is this related to suspected metabolic disorders, or would it include patients with pulmonary or circulatory compromise as an isolated finding. For example, extremely low birthweight infants are unstable, but would not be suspected of having a genetic syndrome. I am thinking more of the general audience here; it would help them to know “who looks like a good candidate for this testing.”

Response: We have published results of six clinical studies that have characterized infants who benefitted from “ultra-rapid WGS” rather than “rapid WGS”. We have cited those references at the end of this sentence. We have changed “unstable infants” to “critically ill infants and children or those with rapid clinical progression in ICUs and who have diseases of unknown etiology^{3-5,7,20,27}.”

9. For the thiamine-responsive seizures case, there are very few details about the case (“At six months of age, he was thriving”). I suggest saying something more objective about the outcome such as something regarding developmental milestones being met, etc.

Response: This patient is now 13.5 months old. We have changed this sentence in the Results as follows: “At thirteen months of age, he has had no further seizures. He is making developmental progress but has delays in gross motor, fine motor and language development.”

Minor specific edits.

(Table S1 and S2) Since this article is directed at more of a general audience, I suggest putting “n/a” in the column for variant 2 when the condition is autosomal dominant. This is obvious to a geneticist, but may not be to a general audience. Alternatively, just explain this in the footnotes of the table.

Response: We have added “n/a” in the column for variant 2 when the condition is autosomal dominant.

(Table S6) is labelled as "Table 6"

Response: We have corrected this.

Reviewer #3 (Remarks to the Author):

In a study, entitled " Genome-to-Treatment: A virtual, automated system for population-scale diagnosis and acute management guidance for genetic diseases in 13.5 hours ", Dr. Owen and colleagues outline the important role of rapid whole genome sequencing with an automated analysis pipeline to establish rapid diagnosis and treatment of rare genetic diseases that have effective treatments as many of those progress rapidly to severe morbidity or mortality if not addressed immediately. They emphasize that front-line physicians are often unfamiliar with these diseases or treatments, hence the need to establish a workflow to follow. The authors describe Genome-to-Treatment (GTRx), an automated, virtual system for genetic disease diagnosis and acute management guidance for ill children in intensive care units. They present examples where diagnosis was achieved in 13.5 hours by sequencing library preparation directly from blood, faster whole genome sequencing (WGS) and informatic analysis, natural language processing of electronic health records and automated interpretation. Upon literature review, they identified 563 severe genetic diseases with effective treatments (drugs, devices, diets, and surgeries) based on clinician nomination by 5 experts in the context of their WGS experience.

The team agreed upon 189 of the first 190 treatments proposed. The authors integrated 10 genetic disease information resources, and electronically linked them and the adjudicated treatments to each automated diagnostic result (<http://gtrx.rbsapp.net/>). This system had superior analytic performance for single nucleotide, insertion-deletion, structural and copy number variants and the author present correct diagnoses and acute management guidance in four retrospective patients. Prospectively, an infant with encephalopathy was diagnosed in 13.5 hours, received effective treatment immediately, and had a good outcome. The authors conclude that GTRx will facilitate broad implementation of optimal acute treatment for children with rapidly progressive genetic diseases by front-line intensive care unit physicians. While this study present an impressive and effective accomplishments by the team in rapidly uncovering genetic variants and linking them to existing therapies and as such of potential interest to the readership of Nature Communications, a significant component of the manuscript involves mining of existing data with recommendations and treatment guidelines that are more in keeping with a review process and this reviewer wonders if the work would not be more appropriate with some modification for a high impact review paper (such as Nat Rev Genet). I have outlined my concerns in the comments below:

Major Comments:

1. The authors generated sequence data in an efficient way and use commercial data mining tools and algorithms (InVitae MOON, Fabric GEMS and Illumina TruSight Software Suite, (TSS)), to annotate the resulting variants in search for a causative variant that confirms diagnosis and would dictate potential therapy. This is of course what numerous genome centers and institutes do on a regular basis and would be helpful if the authors would delineate more specifically what the innovative component is that is unique to their GTRx.

Response: The methods described in this study are unique with regard to time-to-result, scalability, automation, and deliverable (provision of virtual management guidance in addition to a diagnostic report). The following components are innovative and have not been reported previously: 1. WGS library preparation directly from blood and in 70 minutes; 2. WGS in 11 hours; 3. Integration of high performance SV and CNV calling with nucleotide variant calling; 4. Prospective performance of the GEM system for automated interpretation; 5. Development of the Genome-to-Treatment management guidance system; 6. Integration of all of these components in a working prototype. We have revised Figure 1 to highlight the innovative components of GTRx.

2. Similarly, all the data mining work for the known 563 severe genetic diseases with effective treatments is independent of any new data being generated and the innovative contribution here should be brought better to light and avoid commercial influence.

Response: *The Genome-to-Treatment management guidance system will be provided freely to clinicians. The curation of interventions for these genetic diseases is new. Hitherto The vast majority of these interventions had not previously been adjudicated by experts.*

3. It would seem important to address the gap between almost 6000 known diseases and the roughly 500 that have potential therapies – what are the authors doing about the 5500 diseases that have no therapies and what are they recommending for this unmet need.

Response: *We have added the following paragraph to the Discussion:*

“Version 1 of GTRx, described herein, was limited to genetic diseases of known molecular cause, that can be diagnosed by rWGS, can lead to ICU admission in infancy, and have effective treatments. During development, we realized that not all genetic diseases that meet these criteria were represented in the set of 563. Furthermore, the literature related to known genetic diseases and treatments is continually being augmented. While pediatric geneticists were optimal subspecialists for initial review of disorders and interventions, there are many that would benefit from additional sub- and super-specialist review. We plan to address these limitations in future versions of GTRx, with ongoing, expert, open, community-based review. In addition, recent evidence supports the use of rWGS for genetic disease diagnosis and management guidance in older children in PICUs. It is desirable to include these conditions in future versions. There are several, additional, complementary information resources that would enrich GTRx, such as ClinGen, the Genetic Test Registry, and Rx-Genes⁸⁵⁻⁸⁷. Finally, there are many clinical trials of new interventions for infant-onset, severe genetic disorders, particularly genetic therapies. For disorders without current effective treatments it is desirable to include links to enrollment contacts for those clinical trials.”

4. The authors emphasize the strong evidence that exist in support of the notion that diagnosis of genetic diseases by rWGS improves outcomes of infants and children in intensive care units and the approach has been implemented in several countries and multiple states in the US, which this reviewer endorses. The key issue is how best to convey this message to medical practices that are not currently implementing rWGS for their neonatal care, hence the suggestion of a white paper or high impact review manuscript in journals such as Nat Rev Genet.

Response: *We are currently writing a review of the clinical experience with rapid, diagnostic WGS in this population for inclusion in Volume 23 of the Annual Review of Genomics and Human Genetics.*

5. The authors state that another innovation of the automated system they describe is the ability to diagnose genetic diseases associated with all major classes of genomic variants. This is perhaps a bit of an overstatement as is common practice so not terribly innovative and perhaps a different wording should be used to describe this function.

Response: *We have changed “all major classes” to “most major classes”. It is, however, not common practice to diagnose genetic diseases associated with most major classes of genomic variants, such as uniparental isodisomy, solutions for loci with tandem duplications or pseudogenes (such as SNM1 and CYP2D6), triplet repeat expansions, detection of heteroplasmy.*

Minor comments:

1. Important to attenuate any tone driving commercialization of this approach and focus on the science.

Response: *We have edited the manuscript to remove any tone that might be considered to be driving commercialization. We removed company names and logos from Figures 1 and 2 and company names from the associated legends.*

REVIEWERS' COMMENTS

Reviewer #1 (Remarks to the Author):

Thanks for addressing the comments. I have no further question. Hope to see more such implementations.

Reviewer #2 (Remarks to the Author):

As noted previously, this work is likely to be of significant interest in the medical genetics, neonatology, and pediatrics communities in general. The authors have provided the requested clarifications.

Reviewer #3 (Remarks to the Author):

the authors have been responsive to the critique raised by the reviewers and made substantive revisions that that have significantly improved the quality of the manuscript and attenuated any commercial tone. I'm happy with the content of the current revised manuscript.